# Impacts of meteorology and emission reductions on haze pollution during the lockdown in the North China Plain

Lang Liu[1,2], Xin Long[3,*] Yi Li[1,2*], Zengliang Zang[1,2], Fengwen Wang[4], Yan Han[3], Zhier Bao[3], Yang Chen[3], Tian Feng[5], Jinxin Yang[6]

[1]College of Meteorology and Oceanography, National University of Defense Technology, Changsha, 410073, China

[2]Key Laboratory of High Impact Weather (special), China Meteorological Administration, Changsha, 410073, China

[3]Research Center for Atmospheric Environment, Chongqing Institute of Green and Intelligent Technology, Chinese Academy of Sciences, Chongqing, 400714, China

[4]Key Laboratory of the Three Gorges Reservoir Region's Eco-Environment, Ministry of Education, College of Environment and Ecology, Chongqing University, Chongqing, 400030, China

[5]Department of Geography & Spatial Information Techniques, Ningbo University, Ningbo, 315211, China

[6]School of Geography and Remote Sensing, Guangzhou University, Guangzhou, 510006, China

*Correspondence to*: longxin@cigit.ac.cn, liyiqxxy@163.com

**Abstract.**

Haze events in the North China Plain (NCP) during the COVID-19 lockdown underscore the intricate challenges of air quality management amid reduced human activities. Utilizing the WRF-Chem model, we explored how sharp emission reductions and varying meteorological conditions influenced Fine particulate matter ($PM_{2.5}$) concentrations across the NCP. Our analysis highlights a marked regional contrast: in the Northern NCP (NNCP), adverse meteorology largely offset emission reductions, resulting in $PM_{2.5}$ increases of 30 to 60 $\mu g\ m^{-3}$ during haze episodes. Conversely, the Southern NCP (SNCP) benefited from favourable meteorological conditions that lowered $PM_{2.5}$ by 20 to 40 $\mu g$ m-3, combined with emission reductions. These findings emphasize the critical role of meteorology in shaping the air quality response to emission changes, particularly in regions like the NNCP, where unfavourable weather patterns can counteract the benefits of emission reductions. Our study provides valuable insights into the complex interplay of emissions, meteorology, and pollutant dynamics, suggesting that adequate air quality strategies must integrate emissions controls and meteorological considerations to address regional variations effectively.

# 1 Introduction

Fine particulate matter ($PM_{2.5}$) is a critical issue for both policymakers and the general public due to its widespread presence and adverse impacts on human health(Lelieveld et al., 2018), agriculture productivity(Dong and Wang, 2023), and the Earth's radiation balance (Li et al., 2022; Yang et al., 2021). The formation and accumulation of anthropogenic $PM_{2.5}$ result from a complex interaction of emission sources, atmospheric chemical processes, and meteorological conditions (Le et al., 2020). Beyond significant local primary emissions and secondary chemical formation, stagnant meteorological conditions and regional transport significantly contribute to severe haze pollution events (Feng et al., 2020; Li et al., 2021). Since implementing air quality regulations, China has dramatically reduced anthropogenic emissions, leading to a notable decline in $PM_{2.5}$ levels and overall improvements in air quality (Xiao et al., 2020; Zhang et al., 2019). For instance, the Beijing-Tianjin-Hebei (BTH) region witnessed a decline in the number of days with severe $PM_{2.5}$ pollution from 122 days in 2013 to 31 days in 2017 (Li et al., 2019). Despite these improvements, severe $PM_{2.5}$ pollution events still occur. Research has demonstrated that adverse meteorological conditions often play a dominant role in influencing $PM_{2.5}$ concentrations in North China (Le et al., 2020; Shen et al., 2024; Wang et al., 2020), frequently offsetting the positive effects of emission reductions.

The coronavirus disease 2019 (COVID-19) pandemic, which has persisted for over 4.5 years, resulted in more than 7 million deaths globally by June 2023(WHO, 2024). In response to the initial outbreak, the Chinese government enforced stringent lockdowns nationwide during the first 2 months of 2020 to limit the virus's spread (Le et al., 2020). These measures led to a sharp decline in anthropogenic emissions, particularly from the transportation sector (Liu et al., 2021; Xu et al., 2020a). However, during the period from January 21 to February 16, 2020, the Northern China Plain (NCP) experienced severe haze pollution, a stark contrast to other regions (Huang et al., 2021; Le et al., 2020; Wang et al., 2021). This unusual event on the NCP, occurring during a time of reduced human activity, provides a unique opportunity to study the complex interactions between atmospheric chemistry and meteorology under these exceptional conditions.

Recent research on the above haze event in China has highlighted that the unexpected regional haze formation during the COVID-19 lockdown was primarily driven by complex atmospheric

chemical processes influenced by both emission reductions and meteorological factors(Ding et al., 2021; Li et al., 2021). Specifically, the sharp decline in $NO_2$ emissions during the lockdown led to elevated $O_3$ levels and increased night-time formation of $NO_3$ radicals, which boosted the atmospheric oxidation capacity and promoted the generation of secondary aerosols. Furthermore, anomalously high relative humidity during this period facilitated heterogeneous chemical reactions, further contributing to aerosol formation (Huang et al., 2021; Le et al., 2020; Ma et al., 2022). Once formed, these secondary aerosols were transported to monitoring stations in northern China, exacerbating local pollution levels (Lv et al., 2020). Some studies have emphasized that elevated ambient humidity is crucial in enhancing nitrate aerosols' formation efficiency—a key haze component—by influencing pH levels (Chang et al., 2020; Sun et al., 2020). In addition to these chemical interactions, the aerosol–planetary boundary layer (PBL) feedback mechanism is also believed to have significantly contributed to the haze event (Su et al., 2020). Overall, meteorological conditions influenced the formation, accumulation, and dispersion of $PM_{2.5}$ during this period. However, the precise interactions between air pollutants, atmospheric chemistry, and their responses to emissions and meteorological conditions have not been determined.

In this study, we utilized the WRF-Chem model to evaluate the effects of meteorological conditions and abrupt reductions in anthropogenic emissions on $PM_{2.5}$ levels in the NCP. We emphasize the localized differences in how meteorological conditions and emission reductions affect air quality within the North China Plain, specifically between the Northern North China Plain (NNCP) and Southern North China Plain (SNCP). Utilizing the WRF-Chem model, we conducted detailed sensitivity experiments that allowed us to isolate and quantify the individual and combined impacts of emissions and meteorology on air quality, which can deepen the understanding of air quality dynamics in different regional contexts. We addressed three critical questions by simulating severe air pollution episodes during the COVID-19 lockdown: (1) How do sudden emission reductions affect $PM_{2.5}$ levels under varying meteorological scenarios? (2) What are the critical drivers of $PM_{2.5}$ formation and accumulation during these emission reductions? (3) How do meteorological conditions interact with lowered emissions to shape air quality outcomes? Through this analysis, we aim to offer valuable insights into the effectiveness of short-term emission control strategies and to explore the implications of future low-

emission scenarios by examining the combined effects of meteorological variations and emission
reductions on $PM_{2.5}$ concentrations.
**2 Data and methods**
**2.1 Data Sets**
The NCP encompasses 11 provinces and municipalities. This study focused on two sub-regions:
the NNCP and the SNCP. We defined these regions by thoroughly analyzing geographical features,
weather conditions, and emission sources. The NNCP, which generally includes the cities in the
Beijing-Tianjin-Hebei (BTH) area, is surrounded by mountains and elevated terrain to the north and
west. These features make it harder for pollutants to disperse, leading to pollutant buildup, especially in
winter when stagnant atmospheric conditions dominate (Feng et al., 2020; Li et al., 2019). On the other
hand, the SNCP is characterized by lower elevations and broad plains, which help disperse pollutants
due to more vital wind patterns and higher planetary boundary layer heights (Huang et al., 2021). The
emissions in these two regions also differ significantly. The NNCP is mainly affected by concentrated
urban and industrial emissions from the BTH area. At the same time, the SNCP has a broader variety of
sources, including industrial and agricultural emissions, creating a more diverse pollutant profile(Zheng
et al., 2021). These differences in geography, weather, and emissions provide a basis for studying how
meteorological factors and emission reductions affect air quality differently across the NCP (**Figure 1**).
By examining these sub-regions separately, we can better understand how air quality interventions vary
in effectiveness across different areas.
We used two types of air quality data in this study. The first dataset consists of hourly air quality
data provided by the Ministry of Ecology and Environment of China, which has been available since
2013. This dataset includes hourly $PM_{2.5}$, $O_3$, $NO_2$, $SO_2$, and CO concentrations from 823 national
monitoring sites across 185 cities in the study area. Specifically, the NNCP contains 10 cities with 65
measurement sites, while the SNCP includes 24 cities with 95 sampling sites (**Figure 1**). The second
dataset includes chemical compositions such as organic matter, nitrate, sulfate, and ammonium,
collected at the Institute of Atmospheric Physics (IAP), Chinese Academy of Sciences in Beijing, China

(39°58′28″ N, 116°22′16″ E). Detailed descriptions of the methods used to gather these chemical composition data are available in Sun et al. (2020).

We used the Multi-resolution Emission Inventory for China (MEIC), developed by Tsinghua University, with 2016 as the base year (http://meicmodel.org). This emission inventory includes emissions from power plants, transportation, industry, agriculture, and residential activities, with data available at a monthly time scale and a spatial resolution of 6 km. We updated the MEIC inventory to reflect the total provincial emissions estimated for 2020, using near-real-time estimation (Zheng et al., 2021). While the total emissions for each province were updated, the spatial distribution of emissions within each province still followed the intensity proportions from the 2016 MEIC inventory. Subsequently, we applied a top-down approach to adjust further the emission inventory, iteratively comparing model simulations with observed data to refine the estimates until the simulations closely matched the observations. We validated the final emission inventory using statistical parameters, including normalized mean bias (*NMB*), index of agreement (*IOA*), and correlation coefficient (*r*) (**Text S1**). The simulated concentrations were first sampled at each observational site within the region. These site-specific concentrations were then averaged to calculate the regional mean for the NNCP and SNCP, respectively.

The spatial distribution of primary particles ($PM_{2.5}$) and gaseous pollutants ($CO$, $SO_2$, $NO_x$, $NH_3$, and $HCHO$) reveals significantly elevated emission levels across both the NNCP and the SNCP, particularly when compared to the less industrialized northwestern regions of the study area (**Figure S1**). These elevated emissions are primarily driven by dense urbanization and significant industrial activity (Zheng et al., 2021). The topographical features of the NCP, with higher elevations in the north and lower elevations in the south (**Figure 1**), along with substantial pollutant emissions from southern regions, indicate that under persistent southerly winds, pollutants are efficiently transported northward. This northward movement exacerbates air quality degradation, contributing to severe haze episodes in the NNCP, intensifying regional air quality challenges, and complicating mitigation efforts(Huang et al., 2021).

## 2.2 WRF-Chem Model Configuration and Experiments

We employed a specific version (version 3.5.1) of the WRF-Chem model (Grell et al., 2005). We chose the WRF-Chem model because it can simulate coupled atmospheric processes, including emissions, transport, chemical transformations, and aerosol-cloud interactions. This "online" approach allows for dynamic feedback between meteorological conditions and air pollutants. It is well-suited for assessing the interplay between emission reductions and meteorology on $PM_{2.5}$ concentrations during the COVID-19 lockdown period. The model's ability to simultaneously simulate meteorology and chemistry provides advantages over models that treat these processes separately, ensuring that interactions such as aerosol-radiation and aerosol-cloud effects are effectively captured (Li et al., 2011).

Further details regarding the model settings, initial and lateral meteorological and chemical fields, and anthropogenic and biogenic emission inventory(**Table S1**). We used physical schemes of the WRF single-moment(WSM) 6-class graupel microphysical scheme(Hong and Lim, 2006), the Mellor–Yamada–Janjic (MYJ) turbulent kinetic energy planetary boundary layer scheme (Janić, 2001), the unified Noah land-surface model (Chen and Dudhia, 2001) and the Monin-Obukhov surface layer scheme (Janić, 2001). Chemical schemes include the CMAQ/Models-3 aerosol module (Binkowski and Roselle, 2003). Gas-phase reactions of volatile organic compounds (VOCs) and nitrogen oxide ($NO_x$) use the Statewide Air Pollution Research Center-version 1999 (SAPRC99) chemical mechanism. Furthermore, it includes effects such as organic coating on nitrate formation by suppressing the $N_2O_5$ heterogeneous hydrolysis uptake(Liu et al., 2020b), the reaction of stabilized Criegee Intermediates (sCI) with $SO_2$ to form sulfate (Mauldin Iii et al., 2012), and a parameterization of sulfate heterogeneous formation from $SO_2$ involving $Fe^{3+}$ catalyzed and irreversible uptake on aerosol liquid water surfaces (Li et al., 2017a). The Fast Tropospheric Ultraviolet and Visible (FTUV) radiation module calculates photolysis rates, and the model considers the interaction between aerosols and clouds (Li et al., 2011; Tie et al., 2003).

The simulation domain, centered at (116 °E, 38 °N), consisted of 300 × 300 horizontal grid cells with a 6 km resolution (**Figure 1**). The vertical resolution consisted of 35 levels, extending from the surface to 50 hPa, allowing for a detailed representation of boundary layer processes and pollutant dispersion. The initial and boundary meteorological conditions were derived from the National Centers

for Environmental Prediction (NCEP) Final (FNL) reanalysis data at a $1° \times 1°$ spatial resolution and six-hour temporal intervals (Kalnay et al., 2018). Chemical initial and boundary conditions were interpolated from the CAM-Chem (Community Atmosphere Model with Chemistry) global chemistry model(Danabasoglu et al., 2020). The anthropogenic emissions inventory for 2020 was based on a bottom-up approach, incorporating near-real-time data (Zheng et al., 2021), and biogenic emissions were computed online using the Model of Emissions of Gases and Aerosols from Nature (MEGAN)(Guenther et al., 2006). For the episode simulations, the spin-up time is 3 days.

We designed four groups of numerical experiments described in detail in **Table 1**. The first group is the baseline simulation, referred to as the BASE case, covering the period from January 21 to February 16, 2020. This simulation incorporates actual emissions and meteorological conditions during the COVID-19 lockdown period. The BASE case is characterized by reduced emissions, reflecting the unique environmental dynamics during the lockdown.

To quantify the influence of SNCP emissions on $PM_{2.5}$ concentrations in NNCP, we also performed an additional sensitivity test (SNCP0) by setting SNCP emissions to zero within the BASE scenario. The other three groups are sensitivity simulations, which include the emission condition-sensitive simulation (EMIS), the meteorology condition-sensitive simulation (METEO), and the combined emission and meteorology condition-sensitive simulation (EMIS_METEO). In the EMIS experiment, we used the anthropogenic emission inventory from the BASE case. Still, we excluded any abrupt decreases associated with anthropogenic emission reductions during the COVID-19 lockdown period in 2020, following the provincial emission reduction ratios provided by Huang et al. (2021) (**Table S2**). In the METEO case, we applied the same emission inventory as the BASE case but with averaged meteorological conditions from 2015 to 2019. These mean meteorological fields were derived by averaging key meteorological variables (**Text S2**). For the EMIS_METEO case, we used the emission inventory from the EMIS case and the mean meteorological conditions from the METEO case.

The comparison between the BASE and EMIS cases allowed us to evaluate the impact of sudden reductions in anthropogenic emissions on $PM_{2.5}$ levels. The comparison between the BASE and METEO cases provided a stable reference point by reducing the influence of anomalies or fluctuations in meteorological conditions from any year, enabling a comprehensive evaluation of the effects of

meteorological factors on PM$_{2.5}$ levels. Finally, comparing the BASE and EMIS_METEO cases enabled
a thorough assessment of the combined impact of emission reductions and meteorological conditions on
PM$_{2.5}$ levels. Additionally, we analyzed the coupled effects between emission reductions and
meteorological factors using a factor separation approach (**Text S3**).

## 3 Results and Discussions

### 3.1 Model performance

The temporal consistency between model simulations and observations is assessed using *NMB*
and *IOA* (**Table 2 and Figures S2 and S3**). For PM$_{2.5}$ simulations, the average concentration in the
NCP closely matched observations, with an *NMB* of −5.6% and an *IOA* of 0.91 in the NNCP, an *NMB*
of −2.1%, and an *IOA* of 0.86 in the SNCP. For gaseous pollutants, such as SO$_2$, O$_3$, NO$_2$, and CO, the
model effectively captured their diurnal concentration profiles in the NCP region, with *IOA*s exceeding
0.82 in the NNCP and 0.76 in the SNCP. The *NMB*s for these gaseous pollutants also agreed with
observations, with *IOA*s remaining below 6% in the NNCP and below 12% in the SNCP.
The simulated mass concentrations of PM$_{2.5}$ components, including organic matter, nitrate,
sulfate, and ammonium, at the IAP monitoring site, also effectively reproduced the temporal profiles of
these chemical components, with *IOA*s exceeding 0.81. The model shows good agreement with organic
matter and nitrate observations at the IAP observation site, with *NMB*s of 15.0% and −18.9%,
respectively, and *IOA*s exceeding 0.84. However, sulfate is significantly underestimated, with an *NMB*
of −37.7%, which may be attributed to the model's incomplete representation of SO$_2$ oxidation
pathways, particularly through heterogeneous chemistry during haze events(Zheng et al., 2015), and the
acidic aerosol environment (Guo et al., 2017; Liu et al., 2017). Since SO$_2$, as a precursor of sulfate
aerosols, is primarily emitted from point sources, such as power plants or industrial zones, its transport
to observation sites is highly sensitive to uncertainties in wind field simulations, leading to substantial
fluctuations in simulated SO$_2$ and resultant sulfate aerosols. This underestimation in sulfate also affects
ammonium concentrations (NMB = −23.6%) due to its close association with sulfate and nitrate. On a
regional scale, the model's good performance in SO$_2$ simulation (NMB = 4.8% in the NNCP) does not
entirely explain the sulfate underprediction, particularly near the IAP site, where local $SO_2$ is
underestimated by $-12.1\%$ (**Figure S4**). This local discrepancy suggests that WRF-Chem may
inadequately capture oxidation processes such as aqueous-phase and metal-catalyzed reactions, leading
to sulfate underestimation in urban areas with high pollution levels (Guo et al., 2017; Liu et al., 2017;
Zheng et al., 2015). While the model effectively reproduces the temporal variability of critical
components, the consistent underestimation of sulfate and ammonium indicates the need for further
refinements in the representation of $SO_2$ emissions and associated oxidation pathways(Cheng et al.,
2016; Li et al., 2018).
The correlation coefficient indicates the spatial consistency of model simulations compared to
observations (**Figure 2**). During the episode, stagnant meteorological conditions with weak or calm
winds led to unfavorable diffusion of atmospheric pollutants, accumulating and forming heavy haze
pollution in the NCP region. The average simulated $PM_{2.5}$ mass concentrations exceeded 100 µg m$^{-3}$ in
the SNCP and exceeded 120 µg m$^{-3}$ in the NNCP (**Figure 2a**). These results were consistent with
observations, with a correlation coefficient of 0.91 (**Figure 2e**). High $O_3$ levels exceeding 80 µg m$^{-3}$
were simulated over the NNCP region (**Figure 2c**), which indicates an unexpectedly strong atmospheric
oxidation capacity due to weakened titration from low $NO_x$ emissions during the period. During the
episode, almost all avoidable outdoor human activities and most transportation were prohibited. As a
result, the average simulated $NO_2$ (**Figure 2b**) and $SO_2$ (**Figure 2d**) mass concentrations remained very
low in the urban areas of NCP, with values below 30 µg m$^{-3}$ and 10 µg m$^{-3}$, respectively. The spatial
distributions of simulated and observed gaseous pollutants, averaged over the episode, demonstrated
strong spatial consistency, with correlation coefficients ($r$) of 0.67 for $O_3$, 0.86 for $SO_2$, and 0.77 for
$NO_2$ across the research domain (**Figure 2e, 2f**). This high consistency was also observed in the NNCP
and SNCP regions (**Figure S5**), with correlation coefficients for $PM_{2.5}$ and $O_3$ of 0.98 and 0.71 in the
NNCP, and 0.94 and 0.67 in the SNCP. Similarly, the correlation coefficients for $SO_2$ and $NO_2$ were
0.77 and 0.83 in the NNCP, and 0.89 and 0.82 in the SNCP.
The day-to-day variations also show good consistency between the observed and simulated
concentrations of $PM_{2.5}$, $O_3$, $NO_2$, $O_2$, and CO (**Figure 3**). Despite some bias, the WRF-Chem model
captures the temporal and spatial variations of $PM_{2.5}$ and gaseous air pollutants in the BTH region,
which suggests that the emission inventory and simulated meteorological factors are generally
reasonable, providing a reliable basis for further assessment.

## 3.2 Unexpected haze episodes in the NNCP

The COVID-19 pandemic lockdowns in China, which began in late January 2020, led to a sharp
decline in socio-economic activities and a significant reduction in air pollutant emissions (Bao and
Zhang, 2020; Liu et al., 2020a; Wang et al., 2020). In the NNCP, provincial emissions of $NO_x$, $SO_2$, and
$PM_{2.5}$ decreased by 38−45%, 16−26%, and 12−18%, respectively(Huang et al., 2021). Observed
concentrations of $NO_2$ and $SO_2$ significantly decreased to 30.8 µg m$^{-3}$ and 13.5 µg m$^{-3}$, respectively(Li
et al., 2020; Zhao et al., 2020). Satellite observations from the TROPOMI instrument on Sentinel 5P
captured a notable 65% reduction in column-integrated $NO_2$ over eastern China compared to the same
period in 2019(Bauwens et al., 2020; Shi and Brasseur, 2020).
Despite the significant reduction in anthropogenic emissions and lower concentrations of $NO_2$
and $SO_2$, two unexpected heavy haze episodes occurred in the NNCP. Here, we defined haze events as
periods when the daily average $PM_{2.5}$ concentration in the NNCP exceeds 100 µg m$^{-3}$. During the study
period, two significant haze episodes were identified: EP1, lasting from January 22 to 29, and EP2,
from February 8 to 13. During EP1, the average $PM_{2.5}$ concentration in the NNCP reached 153.4 µg m$^{-3}$,
peaking at approximately 185 µg m$^{-3}$, significantly higher than in the SNCP, which peaked at around
120 µg m$^{-3}$. In EP2, the average $PM_{2.5}$ concentration in the NNCP reached 132.2 µg m$^{-3}$, peaking at
approximately 150 µg m$^{-3}$. No haze was observed in SNNP during EP2, with average $PM_{2.5}$
concentrations of 57.7 µg m$^{-3}$ (**Figure 3**).
During EP1, stagnant atmospheric conditions in the NNCP with wind speeds lower than 0.8 m s$^{-1}$
(**Figures 4c, S6b, S6c**), coupled with a low planetary boundary layer height (PBLH) of approximately
306 m (ranging from 190 to 454 m) (**Figure S6a**), facilitated the accumulation of pollutants. Under
these conditions, $PM_{2.5}$ concentrations (**Figure 3a**) reached peak values of around 150−200 µg m$^{-3}$, and
$O_3$ concentrations (**Figure 3b**) steadily increased, peaking at approximately 90 µg m$^{-3}$. This trend
indicates enhanced photochemical activity due to the stagnant conditions. Concurrently, $NO_2$
concentrations (**Figure 3c**) decreased, likely due to its conversion to $O_3$ and secondary aerosols. The
consistently high levels of $SO_2$ and CO (**Figures 3d and 3e**) further indicated the limited dispersion
under static atmospheric conditions. These conditions facilitated photochemical reactions, enhancing
secondary pollution formation, as suggested by recent studies on secondary pollution during the
COVID-19 lockdown(Huang et al., 2021).
In contrast, during EP2, the concentrations of $PM_{2.5}$, $O_3$, $NO_2$, $SO_2$, and CO (**Figure 3**) exhibited
bell-shaped styles fluctuating pattern, performing with the simultaneous increase and decrease of
various pollutants. These fluctuating patterns indicate dynamic atmospheric conditions with significant
air pollutant transport and mixing processes. The northward speeds of about 4.1 m s$^{-1}$ in the SNCP
facilitated the transport of air pollutants from the SNCP to the NNCP(**Figures 4d, S6b**). Simultaneously,
stagnant atmospheric conditions in the NNCP with wind speeds lower than 0.5 m s$^{-1}$, corresponding
with low PBLH of 306 m (ranging from 209 to 458 m) (**Figure S6a**), facilitated the accumulation of
pollutants in the NNCP.
Overall, the contrasting atmospheric conditions during EP1 and EP2 underscore the complex
interplay of meteorological factors and their significant impact on pollutant levels in the NNCP. The
stagnant conditions during EP1 led to significant pollutant accumulation and secondary pollution
formation, while the dynamic conditions during EP2 highlighted the role of regional pollutant transport
in exacerbating haze episodes. These findings emphasize the need to consider local and regional
atmospheric processes in air quality management strategies.
Reducing anthropogenic emissions has been a primary factor in decreasing $PM_{2.5}$ pollution in
China(Bao and Zhang, 2020; Liu et al., 2020a). However, these haze episodes in NNCP during the
COVID-19 lockdown challenge the relationship between human activities and air quality. These
unexpected haze episodes underscore the complexity of air quality dynamics, suggesting that factors
such as meteorological conditions, secondary pollutant formation, regional transport, and non-industrial
sources also significantly impact air quality (Huang et al., 2021; Liu et al., 2020a; Shi and Brasseur,
2020). Future air quality management strategies must incorporate these multifaceted interactions for
more effective pollution control.

### 3.3 Meteorological conditions increase PM$_{2.5}$ in NNCP and decrease it in SNCP

Meteorological factors significantly influenced PM$_{2.5}$ concentrations during the study period, as illustrated by the pattern comparisons between the "BASE" and "METEO" simulations (**Figure 5a**). Changes in PM$_{2.5}$ concentrations ranged from decreases of up to 50 µg m$^{-3}$ to increases exceeding 100 µg m$^{-3}$, revealing an apparent north-south disparity. In the NNCP, meteorological conditions led to significant increases in PM$_{2.5}$ concentrations, particularly in the northern regions, where levels rose by 50 to 100 µg m$^{-3}$. In contrast, the SNCP, especially the western parts, experienced a decrease in PM$_{2.5}$ levels by 30 to 50 µg m$^{-3}$, reflecting the more favorable meteorological conditions that facilitated pollutant dispersion.

During haze episodes (EP1 and EP2), meteorological conditions had an even more pronounced effect. In EP1, PM$_{2.5}$ concentrations in the NNCP increased by 30 to 100 µg m$^{-3}$ (**Figure 5c**), particularly in the central NNCP areas near the mountain foothills. Meanwhile, the SNCP benefited from reductions in PM$_{2.5}$ concentrations of 30 to 50 µg m$^{-3}$, suggesting that enhanced pollutant dispersion helped mitigate pollution in the southern region. The impact of meteorology was even more substantial during EP2, with PM$_{2.5}$ increases in the NNCP exceeding 100 µg m$^{-3}$ in some areas, and reaching up to 150 µg m$^{-3}$ in heavily affected regions (**Figure 5d**). Low planetary boundary layer heights (PBLH) and stagnant surface winds drove this increase, particularly in Beijing and its surrounding areas (**Figure S7c, S7d**). Conversely, in the SNCP, reductions in PM$_{2.5}$ concentrations of 30 to 50 µg m$^{-3}$ were observed, aided by higher PBLH and stronger northward winds, which enhanced pollutant dispersion. Meanwhile, the comparison between the "SNCP0" simulation (with SNCP emissions set to zero) and the "BASE" case demonstrated a substantial reduction in PM$_{2.5}$ concentrations in the NNCP (**Figure S8**), particularly during EP2. This reduction, ranging from 15 to 30 µg m$^{-3}$ in some regions of the NNCP (**Figure S8b**), provides direct evidence that SNCP emissions contribute significantly to PM$_{2.5}$ accumulation in the NNCP via northward transport. This finding underscores the importance of regional transport, facilitated by northward winds, in elevating PM$_{2.5}$ concentrations in the NNCP, especially under meteorological conditions that support pollutant movement from south to north.

During non-haze periods, weather conditions still significantly impacted PM$_{2.5}$ levels across the region, though the effect was less intense than haze episodes. In the NNCP, stagnant air and low wind speeds led to PM$_{2.5}$ increases of 10 to 30 µg m$^{-3}$ (**Figure 5b**). These weak conditions prevented effective pollutant dispersion, causing pollutants to accumulate, although less than during significant pollution events. This ongoing buildup due to poor weather shows the continued vulnerability of the NNCP to limited ventilation (Feng et al., 2021; Yan et al., 2024). In contrast, in the SNCP, weather conditions helped reduce PM$_{2.5}$ by 10 to 30 µg m$^{-3}$ (**Figure 5b**). This improvement was mainly due to higher PBLH (**Figure S7b**) and stronger winds (**Figure 5b**), which promoted pollutant dispersion. The PBLH rose by 100 to 300 meters, allowing pollutants to spread vertically, leading to lower PM$_{2.5}$ levels at the surface. Favorable winds also helped clear pollutants, enhancing the positive effects of meteorology on air quality. Previous studies have shown that regions with better dispersion conditions can achieve more significant air quality improvements, even with similar emissions, due to more efficient pollutant removal (Xu et al., 2020b; Zhang et al., 2021). These regional differences during non-haze periods show the critical role of weather in influencing air quality. In the NNCP, weak atmospheric circulation limited pollutant dispersion, causing moderate PM$_{2.5}$ increases. In contrast, in the SNCP, more dynamic weather conditions promoted pollutant removal, leading to substantial reductions.

Regional variations in haze episodes underscore the critical role of elevated near-surface temperature (T2) and relative humidity (RH) in driving secondary aerosol formation (**Figure S9**). In the NNCP, elevated T2 accelerates gas-phase oxidation reactions, converting volatile organic compounds (VOCs) and nitrogen oxides (NO$_x$) into secondary organic aerosols (SOAs) and nitrate aerosols, thus contributing to increased PM$_{2.5}$ levels despite reduced emissions (Huang et al., 2021; Seinfeld and Pandis, 2016). Similarly, elevated RH facilitates aqueous-phase reactions that convert SO$_2$ into sulfate on particle surfaces, aided by aerosol liquid water, and this effect is particularly pronounced during haze episodes, where high RH accelerates sulfate formation even with decreased emissions (Le et al., 2020; Wang et al., 2020). The online WRF-Chem model captures these interactions in the SEN_METEO simulation, integrating the effects of T2 and RH into the modeled PM$_{2.5}$ concentrations. Although the study does not isolate each specific chemical pathway, the correlation between elevated T2, RH, and

higher PM$_{2.5}$ concentrations aligns with previous research, and underscores the pivotal role of meteorological conditions in secondary aerosol formation. This finding highlights the importance of considering meteorological influences in addition to emission reductions, as unfavorable weather conditions can offset the expected improvements from reduced emissions and sustain elevated PM$_{2.5}$ levels. This understanding is essential for developing effective air pollution control strategies that account for emissions and meteorological variability.

These meteorological effects also impact secondary aerosols, including secondary organic aerosols (SOAs) and secondary inorganic aerosols (SIAs), with substantial variability between the NNCP and SNCP regions. In the NNCP, stagnant conditions and reduced boundary layer heights limited pollutant dispersion, contributing to the accumulation of SOAs and SIAs. High humidity further exacerbated the formation of secondary aerosols, resulting in elevated concentrations (**Figure S10**). Conversely, the SNCP benefited from higher PBLH (**Figure S7**) and dynamic wind patterns(**Figure 4a**), which enhanced the dispersion of both primary and secondary aerosols, reducing their concentrations. Due to the very low emissions of biogenic secondary organic aerosol (BSOA) precursors during wintertime(Guenther et al., 2012), the BSOA contribution to PM$_{2.5}$ concentrations is insignificant, averaging less than 2 µg m$^{-3}$ throughout the study period (**Figure S11a**). The average BSOA accounted for less than 2% of total PM$_{2.5}$ mass in the BASE simulations (**Figure S11b**), indicating a minor role for biogenic emissions in shaping wintertime air quality.

**3.4 Emission reduction decreases the PM$_{2.5}$ in the NSCP and SNCP**

Abrupt decreases in anthropogenic emissions during the COVID-19 lockdown led to significant reductions in PM$_{2.5}$ concentrations across both the NNCP and SNCP (**Figure 6a**). Both regions experienced substantial PM$_{2.5}$ decreases, contributing to improvements in air quality. In addition to the overall PM$_{2.5}$ reductions, emission controls significantly impacted SOAs and SIAs in the NNCP and SNCP (**Figure S10b, 10d**). The reductions in SOAs and SIAs were driven by decreased availability of precursors such as VOCs for SOAs and SO$_2$ and NO$_x$ for SIAs(Huang et al., 2021).

Wintertime ozone production in urban areas of northern China typically occurs in a NO$_x$-saturated regime, primarily due to a lack of HOx radicals and limited solar radiation during

winter(Seinfeld and Pandis, 2016). Additionally, reduced fresh NO emissions alleviate ozone
titration(Levy et al., 2014). Thus, a reduction in $NO_x$ often leads to increased ozone levels. In the NCP
during winter, there is usually an inverse relationship between $PM_{2.5}$ and $O_3$, attributed to the aerosol
radiative effect on ozone photochemistry(Li et al., 2017b; Wu et al., 2020). However, during the
COVID-19 lockdown, this inverse relationship disappeared in the NNCP, with ozone concentrations
reaching approximately 65.7 μg m$^{-3}$ even when $PM_{2.5}$ levels exceeded 100 μg m$^{-3}$ (**Figure S12**).
Significant reductions in $NO_x$ emissions reduced ozone titration, resulting in elevated ozone levels
despite higher $PM_{2.5}$ concentrations. This pattern aligns with previous findings that in $NO_x$-saturated
environments, reductions in $NO_x$ can increase ozone levels, with additional effects from aerosol
radiative influences and precursor interactions shaping the $O_3-PM_{2.5}$ relationship(Le et al., 2020). These
dynamics highlight the importance of considering nonlinear chemical and meteorological factors when
assessing air quality responses to emission reductions.
During haze episodes (EP1 and EP2), the absolute decrease in $PM_{2.5}$ was considerably greater
than during non-haze periods. $PM_{2.5}$ reductions during these episodes generally exceeded 30 to 50 μg
m$^{-3}$ (**Figure 6c, 6d**), particularly in areas along the mountain foothills, where contributions surpassed 50
μg m$^{-3}$ during EP2 (**Figure 6d**). This considerable decrease underscores the enhanced effectiveness of
emission control measures during severe pollution events, highlighting the importance of emission
reductions in extreme pollution levels(Zheng et al., 2021).
In non-haze periods, the reductions in $PM_{2.5}$ were less pronounced, typically ranging from 5 to
30 μg m$^{-3}$ (**Figure 6b**). These results suggest that emissions reductions effectively lowered $PM_{2.5}$
concentrations, but their impact was more moderate under baseline conditions with lower pollution
levels. The sensitivity simulations confirm that emission reductions during the lockdown directly
contributed to decreased $PM_{2.5}$ levels across regions.
It is important to note that the reductions seen in the EMIS scenario are attributed solely to
changes in emissions and do not account for meteorological influences. The meteorological conditions
during the study period likely offset some emission-driven improvements, which will be further
explored in the combined effects analysis. However, the EMIS results demonstrate the potential

effectiveness of emission controls in reducing $PM_{2.5}$, particularly in regions with high anthropogenic activity.

**3.5 Combined and coupled effects of meteorology and emission reduction on $PM_{2.5}$**

The combined and coupled effects of meteorological conditions and emission reductions during the COVID-19 lockdown significantly influenced $PM_{2.5}$ concentrations in the NNCP and SNCP. These effects varied depending on the region and the interaction between meteorological factors and reduced emissions, aligning with findings from similar studies in urban areas during lockdowns that emphasize the role of meteorology in modulating pollution levels (Huang et al., 2021).

The results highlight contrasting impacts between the NNCP and SNCP regarding combined effects. In the NNCP, the combined effects of weather conditions and emission reductions led to noticeable increases in $PM_{2.5}$ levels during the study period. These combined effects raised $PM_{2.5}$ concentrations by 10 to 75 µg m$^{-3}$, especially in the northern regions (**Figure 7a**). Even during non-haze periods, this combined influence caused $PM_{2.5}$ to increase by 10 to 40 µg m$^{-3}$ (**Figure 7b**). The impact was even more significant during haze episodes. For example, during EP2, $PM_{2.5}$ levels increased by exceeding 100 µg m$^{-3}$ (**Figure 7d**), showing that adverse weather conditions, like stagnant winds and low boundary layer heights, negated the benefits of emission reductions. In the SNCP, the combined effects led to significant decreases in $PM_{2.5}$ levels. Throughout the study period, $PM_{2.5}$ concentrations dropped by 30 to 100 µg m$^{-3}$ (**Figure 7a**). The positive impact of emission reductions was most apparent during haze episodes, where the combined effects during EP2 led to reductions exceeding 100 µg m$^{-3}$ in some areas (**Figure 7d**).

The factor separation analysis provided critical insights into the combined effects of emissions and meteorology (**Figure S13**). During non-haze periods(**Figure S13b**), the coupled effects contributed to a $PM_{2.5}$ increase of 5 to 10 µg m$^{-3}$ in the NNCP. Still, they increased to 10 to 50 µg m$^{-3}$ during haze episodes, particularly during EP2 (**Figure S13d**). This indicates that unfavorable meteorological conditions limited the effectiveness of emission reductions in the NNCP. As a result, emission reductions, though beneficial, were insufficient to improve air quality significantly under these conditions. This finding aligns with previous studies showing that areas with adverse weather

conditions often struggle to improve air quality despite emission reductions (Feng et al., 2021). Such conditions hinder pollutant dispersion, making it difficult for emission reductions to decrease PM2.5 concentrations significantly (Zheng et al., 2021).

In contrast, the SNCP exhibited more vital coupled effects between meteorology and emission reductions. During haze episodes, this interaction led to an additional 10 to 50 $\mu g\ m^{-3}$ reduction in $PM_{2.5}$ levels (**Figure S13c, S13d**). The coupled effects between favorable meteorological conditions and reduced emissions greatly enhanced $PM_{2.5}$ decreases, especially during the EP2 haze episode. This more substantial interaction in the SNCP highlights how favorable meteorology can amplify the impact of emission reductions, leading to more vital improvements in air quality. Previous research has shown that when meteorology supports pollutant dispersion, the benefits of emission reductions are maximized, resulting in significant decreases in pollutant concentrations(Xu et al., 2020b; Zhang et al., 2021).

The station-averaged regional contributions also reveal differences between the NNCP and SNCP during the COVID-19 lockdown (**Figure 8**). In the NNCP, adverse meteorological conditions dominated, driving significant $PM_{2.5}$ increases of 60 to 90 $\mu g\ m^{-3}$ during haze episodes. In comparison, emission reductions contributed more modest decreases of 20 to 40 $\mu g\ m^{-3}$. Coupled effects added only 10 to 15 $\mu g\ m^{-3}$ in reductions, insufficient to offset the impact of poor weather(**Figure 8a**). Conversely, in the SNCP, emission reductions had a more substantial effect, with $PM_{2.5}$ levels decreasing by 30 to 50 $\mu g\ m^{-3}$ during haze episodes, as meteorology and emissions worked synergistically. Coupled effects in the SNCP contributed an additional 15 to 20 $\mu g\ m^{-3}$ in reductions, highlighting a more vital interaction between favorable meteorology and emissions controls (**Figure 8b**). Daily contributions support these trends, with the NNCP seeing persistent increases, while the SNCP experienced consistent reductions, especially during EP2, where daily decreases ranged from 40 to 60 $\mu g\ m^{-3}$ (**Figure S14**).

## 4 Conclusions

This study highlights the significant but regionally variable impacts of meteorological conditions and emission reductions on $PM_{2.5}$ levels across the NCP during the COVID-19 lockdown. In the NNCP, adverse meteorological conditions, characterized by cold, stagnant, and humid air masses, often outweighed the benefits of emission reductions, leading to increased $PM_{2.5}$ concentrations, especially

during haze episodes. Conversely, in the SNCP, warmer air masses and more favourable meteorological
conditions enhanced the effectiveness of emission reductions, resulting in decreased $PM_{2.5}$ levels.
Previous studies have primarily focused on the overall impacts of meteorological conditions and
emission reductions on air quality across the North China Plain and even nationwide. We emphasize the
localized differences in how meteorological conditions and emission reductions affect air quality within
the North China Plain, specifically between the NNCP and SNCP. Our findings underscore the critical
role that meteorological conditions play in modulating the effects of emission reductions. The
combination of unfavourable meteorological factors and emission reductions in the NNCP led to overall
increases in $PM_{2.5}$ levels, with significant increases during haze episodes. Meanwhile, in the SNCP,
meteorological conditions and emission reductions consistently contributed to lower $PM_{2.5}$
concentrations.
These results emphasize the necessity of integrated air quality management strategies for
emission sources and atmospheric dynamics. By understanding the spatial and temporal variations in
$PM_{2.5}$ in response to different meteorological conditions, policymakers can design more effective
pollution control measures, particularly during critical pollution episodes. This study provides valuable
insights into the complex interactions between emissions, meteorology, and air quality, highlighting the
need for comprehensive approaches to improve air quality in the NCP.
*Data availability*
The code and data used in this study are from Lang Liu (liulang@ieecas.cn) and Xin Long
(longxin@cigit.ac.cn).
*Competing interests*
The authors declare that they have no conflict of interest.

*Author contribution*

LL and XL designed the research and wrote the manuscript. YL, ZZ, FW, YY, ZB, TF, and JY contributed to interpreting the results. All the authors provided critical feedback and helped to improve the manuscript.

*Acknowledgments*

This work was supported by the National Natural Science Foundation of China (grant no. 42007206, and U23A2030), the Science and Technology Innovation Program of Hunan Province (2024RC3129), the Fund and Program of National University of Defense Technology (202301-YJRC-ZZ-002, ZK23-52), and the Open Fund of the State Key Laboratory of Loess and Quaternary Geology (grant no. SKLLQG2219). The authors also thank Tsinghua University for compiling and sharing the MEIC.

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

**Figure Captions**

**Figure 1.** The simulation domain in WRF-Chem, including topography. Circles represent the locations of cities with ambient air quality monitoring sites, with circle size reflecting the number of monitoring sites per city. The IAP observation sites are marked with black pentagons. The regions of interest, NNCP (Northern North China Plain) and SNCP (Southern North China Plain), are highlighted.

**Figure 2.** The pattern comparisons between average observations and simulations for (a) $PM_{2.5}$, (b) $SO_2$, (c) $O_3$, and (d) $NO_2$. Additionally, statistical comparisons are presented for (e) $PM_{2.5}$ and $O_3$, and (f) $SO_2$ and $NO_2$, along with their correlation coefficients (*r*).

**Figure 3.** Observed (solid lines) and simulated (dashed lines) day-to-day variations in surface $PM_{2.5}$ $O_3$, $NO_2$, $SO_2$, and CO levels in the NNCP (red lines) and SNCP (blue lines) from January 21 to February 15, 2020. The daily concentrations of the pollutants were calculated from the 24-hour averages, except for $O_3$, which was calculated from the 10:00 to 17:00 averages. Two haze episodes occurred during the study period: EP1 from January 22 to 29, and EP2 from February 8 to 13.

**Figure 4.** The spatial patterns of near-surface simulated $PM_{2.5}$ averaged from (a) the entire study period, (b) the non-haze period, (c) the EP1 haze period, and (d) the EP2 haze period, along with the simulated surface wind fields.

**Figure 5.** The pattern comparisons of the "BASE" simulation minus the "METEO" simulation. The color gradient represents PM2.5 changes averaged from (a) the entire study period, (b) the non-haze period, (c) the EP1 haze period, and (d) the EP2 haze period, along with the simulated surface wind fields.

**Figure 6.** The pattern comparisons of the "BASE" simulation minus the "EMIS" simulation. The color gradient represents $PM_{2.5}$ changes averaged from (a) the entire study period, (b) the non-haze period, (c) the EP1 haze period, and (d) the EP2 haze period.

**Figure 7.** The pattern comparisons of the "BASE" simulation minus the "EMIS_METEO" simulation. The color gradient represents coupled effects on $PM_{2.5}$ averaged from (a) the entire study period, (b) the non-haze period, (c) the EP1 haze period, and (d) the EP2 haze period.

**Figure 8.** Regional contributions to $PM_{2.5}$ averaged in (a) the NNCP and (b) the SNCP during the entire period, non-haze period, EP1, and EP2. The contributions include meteorological conditions (METEO), abrupt anthropogenic emissions (EMIS) decreases, and coupled and combined effects of METEO and EMIS.

**Table Captions**

**Table 1** Configurations of simulation cases in this study

**Table 2.** The statistical parameters of model performance include temporal assessments of *MB*, and *IOA* in the NNCP and SCNP and at the IAP monitoring site.

**Figure 1**

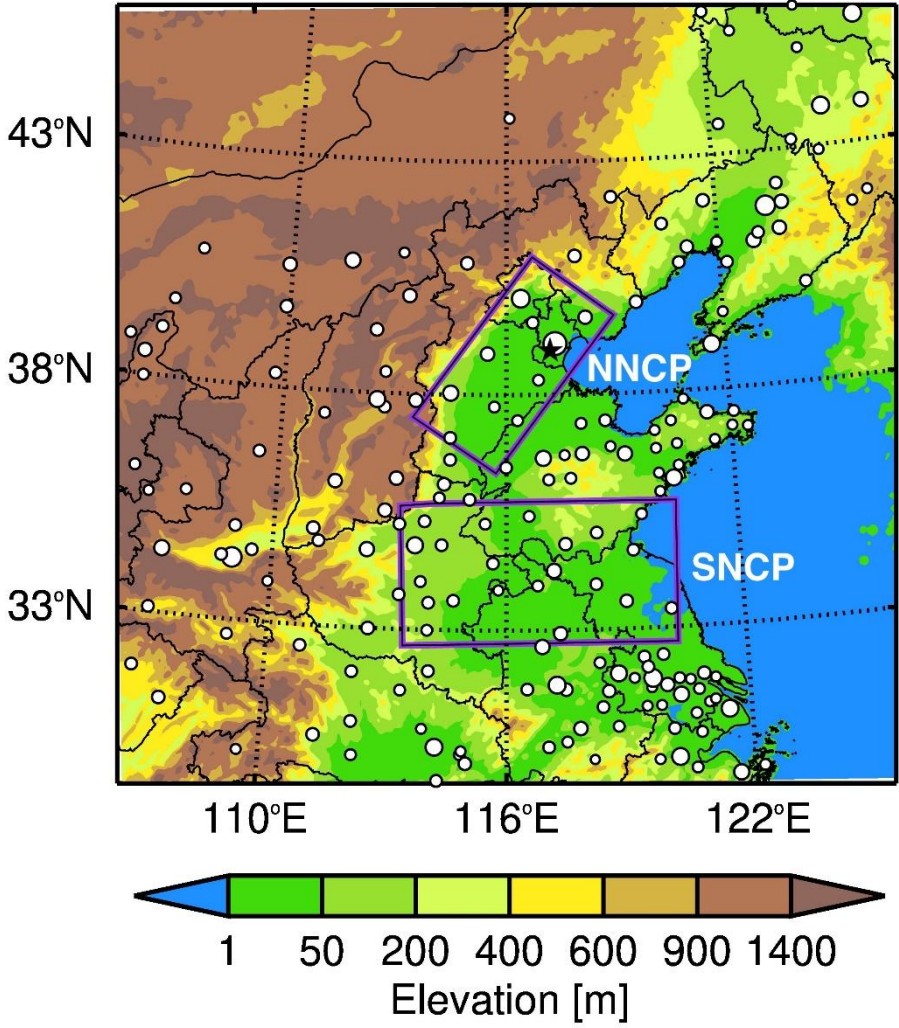


**Figure 1.** The simulation domain in WRF-Chem, including topography. Circles represent the locations of cities
with ambient air quality monitoring sites, with circle size reflecting the number of monitoring sites per city. The
IAP observation sites are marked with black pentagons. The regions of interest, NNCP (Northern North China
Plain) and SNCP (Southern North China Plain), are highlighted.

**Figure 2**

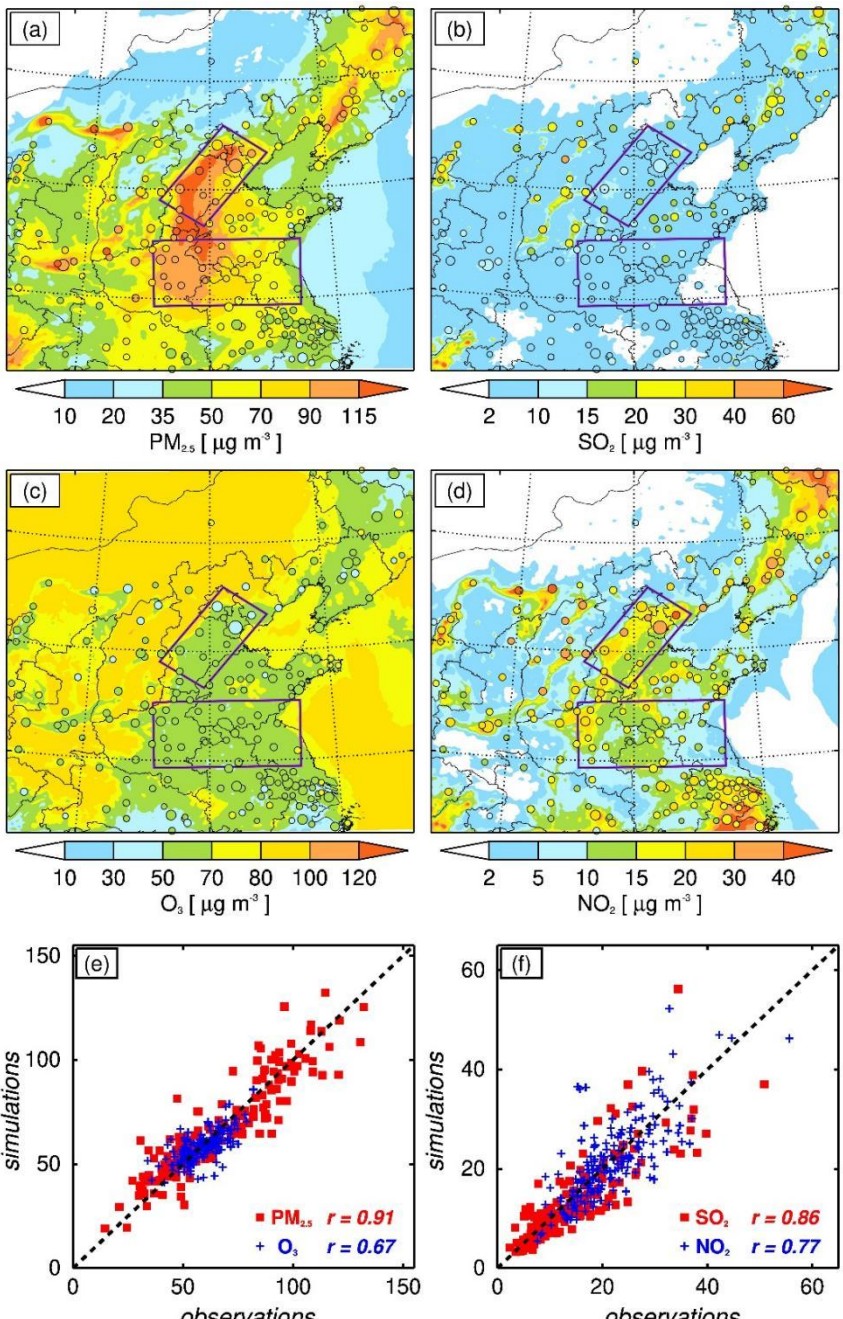


**Figure 2.** The pattern comparisons between average observations and simulations for (a) PM$_{2.5}$, (b) SO$_2$, (c) O$_3$, and (d) NO$_2$.
Additionally, statistical comparisons are presented for (e) PM$_{2.5}$ and O$_3$, and (f) SO$_2$ and NO$_2$, along with their correlation
coefficients ($r$).
**Figure 3**

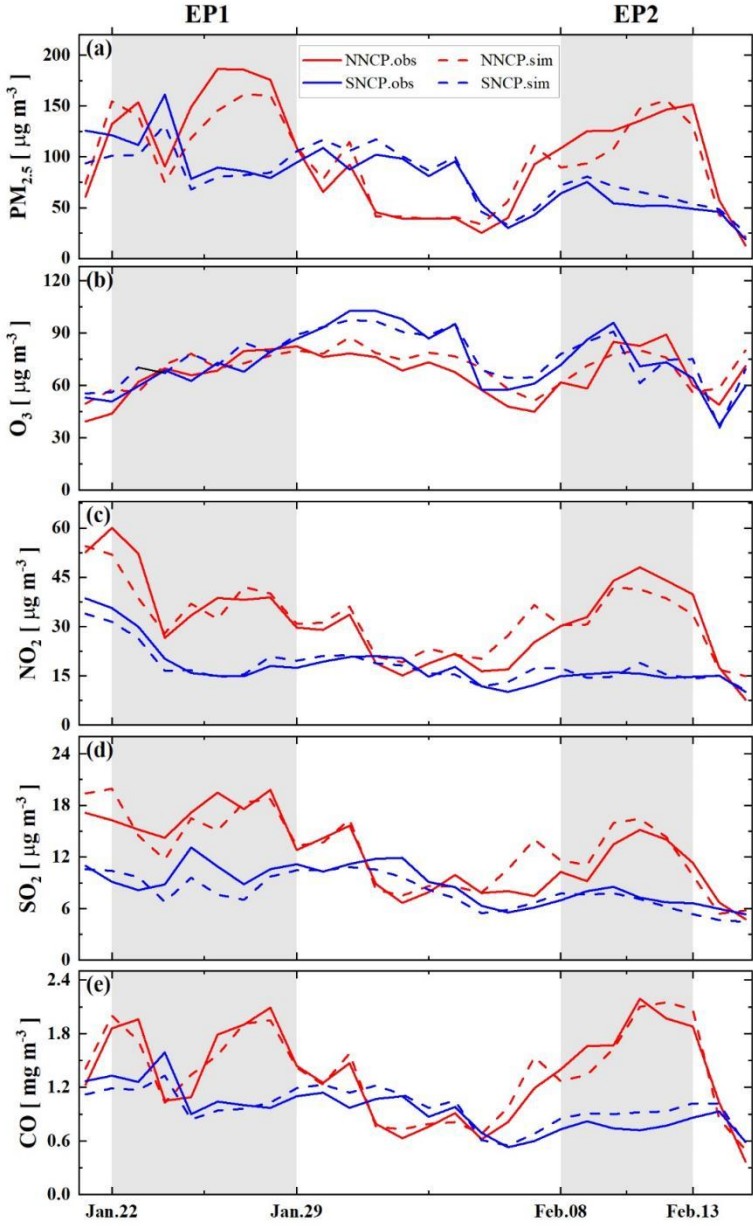


**Figure 3.** Observed (solid lines) and simulated (dashed lines) day-to-day variations in surface PM$_{2.5}$ O$_3$, NO$_2$, SO$_2$, and
CO levels in the NNCP (red lines) and SNCP (blue lines) from January 21 to February 15, 2020. The daily
concentrations of the pollutants were calculated from the 24-hour averages, except for O$_3$, which was calculated from
the 10:00 to 17:00 averages. Two haze episodes occurred during the study period: EP1 from January 22 to 29, and EP2
from February 8 to 13.
**Figure 4**

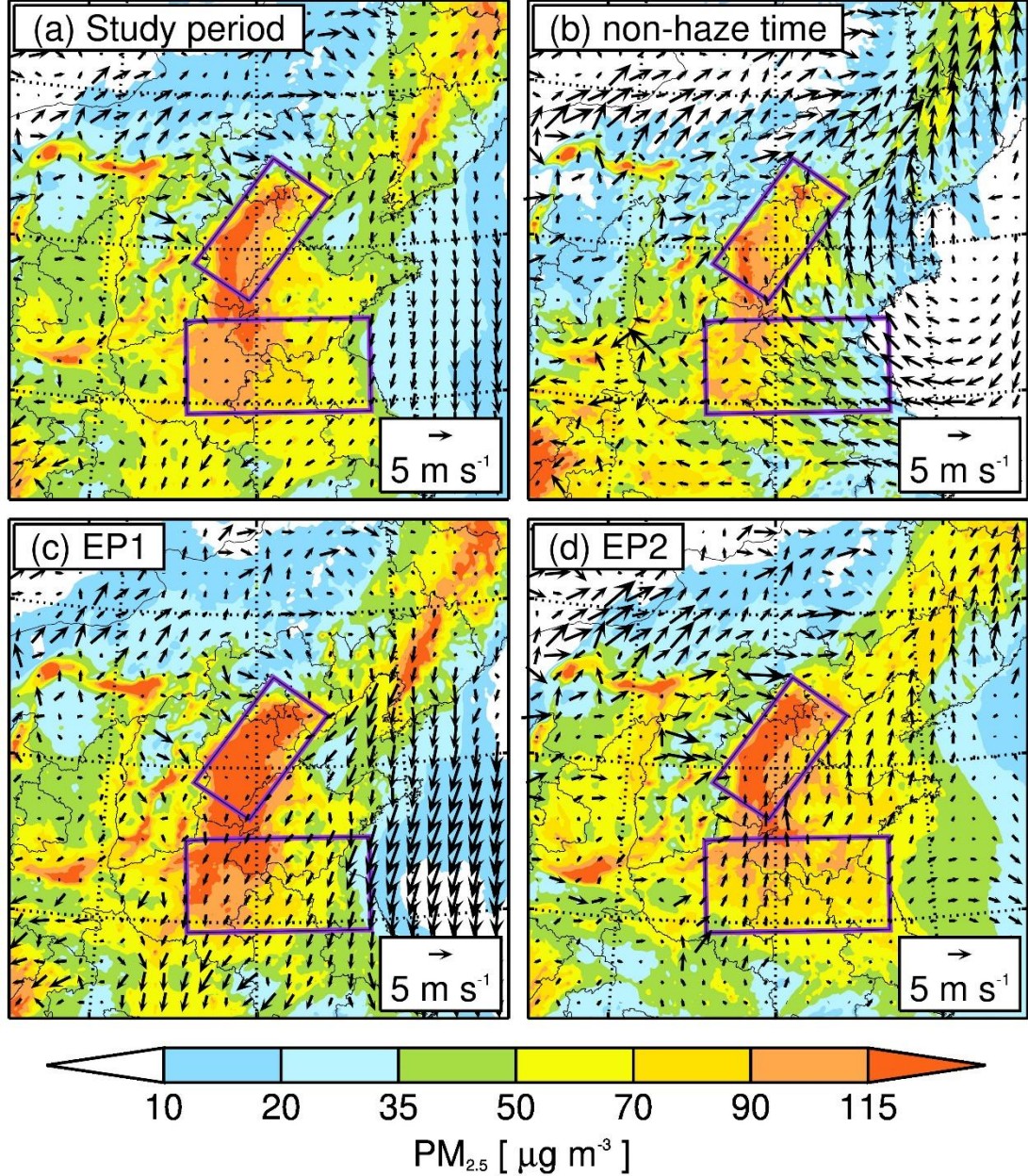

**Figure 4.** The spatial patterns of near-surface simulated PM₂.₅ averaged from (a) the entire study period, (b) the non-haze period, (c) the EP1 haze period, and (d) the EP2 haze period, along with the simulated surface wind fields.


**Figure 4.** The spatial patterns of near-surface simulated PM$_{2.5}$ averaged from (a) the entire study period, (b) the non-haze
period, (c) the EP1 haze period, and (d) the EP2 haze period, along with the simulated surface wind fields.
**Figure 5**

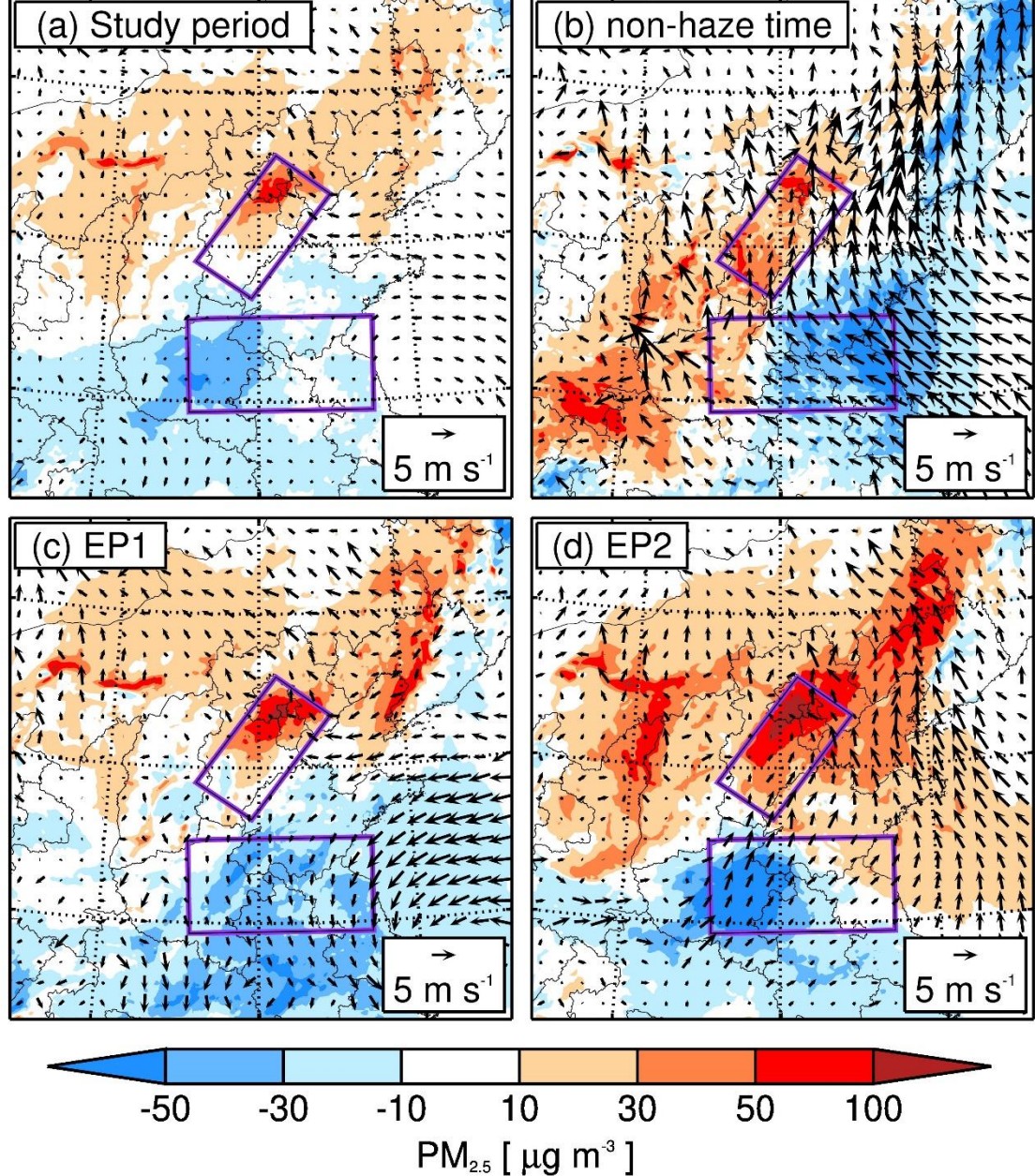

**Figure 5.** The pattern comparisons of the "BASE" simulation minus the "METEO" simulation. The color gradient represents
PM2.5 changes averaged from (a) the entire study period, (b) the non-haze period, (c) the EP1 haze period, and (d) the EP2
haze period, along with the simulated surface wind fields.

**Figure 6**

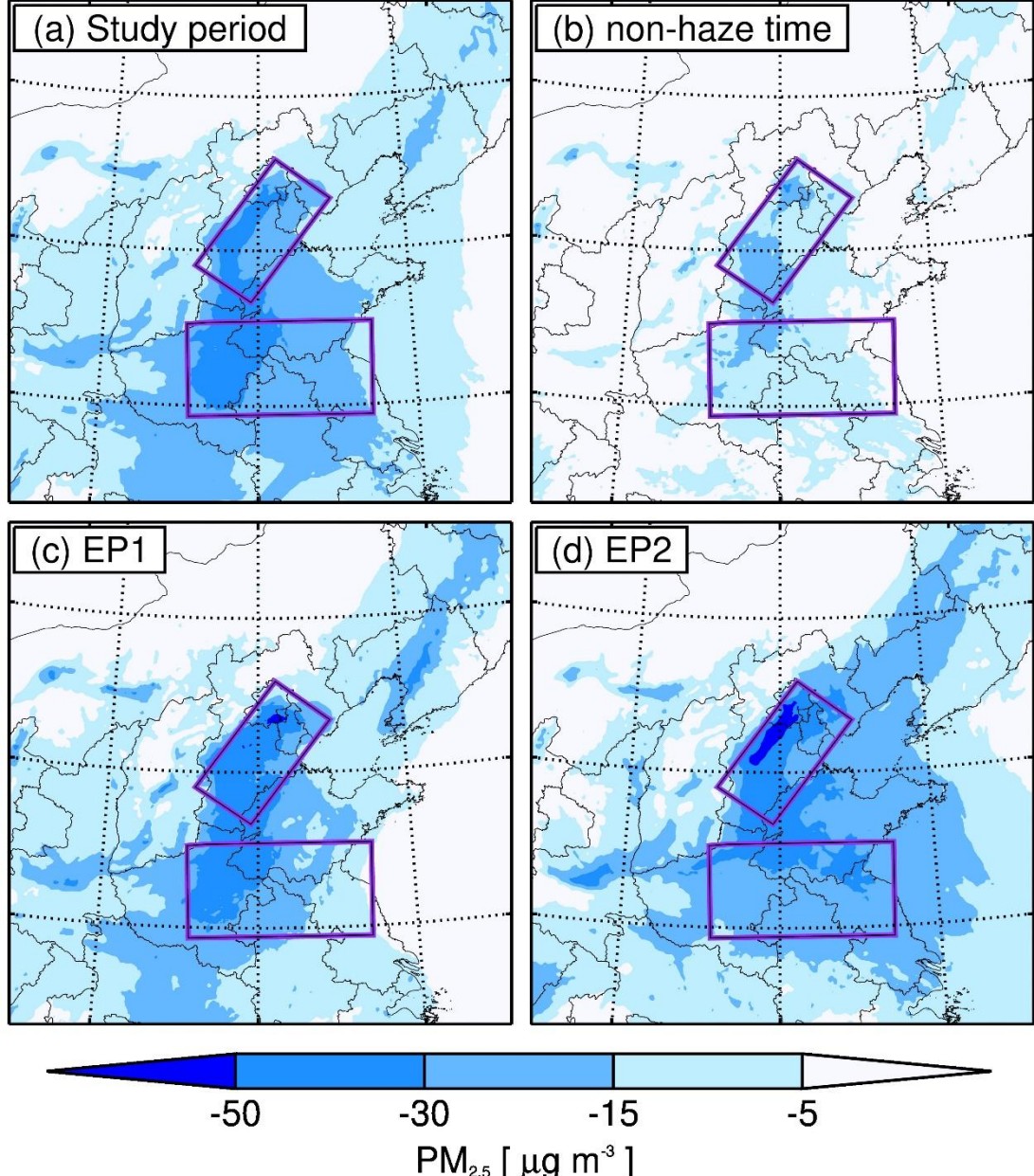


**Figure 6.** The pattern comparisons of the "BASE" simulation minus the "EMIS" simulation. The color gradient represents
PM$_{2.5}$ changes averaged from (a) the entire study period, (b) the non-haze period, (c) the EP1 haze period, and (d) the EP2
haze period.

**Figure 7**

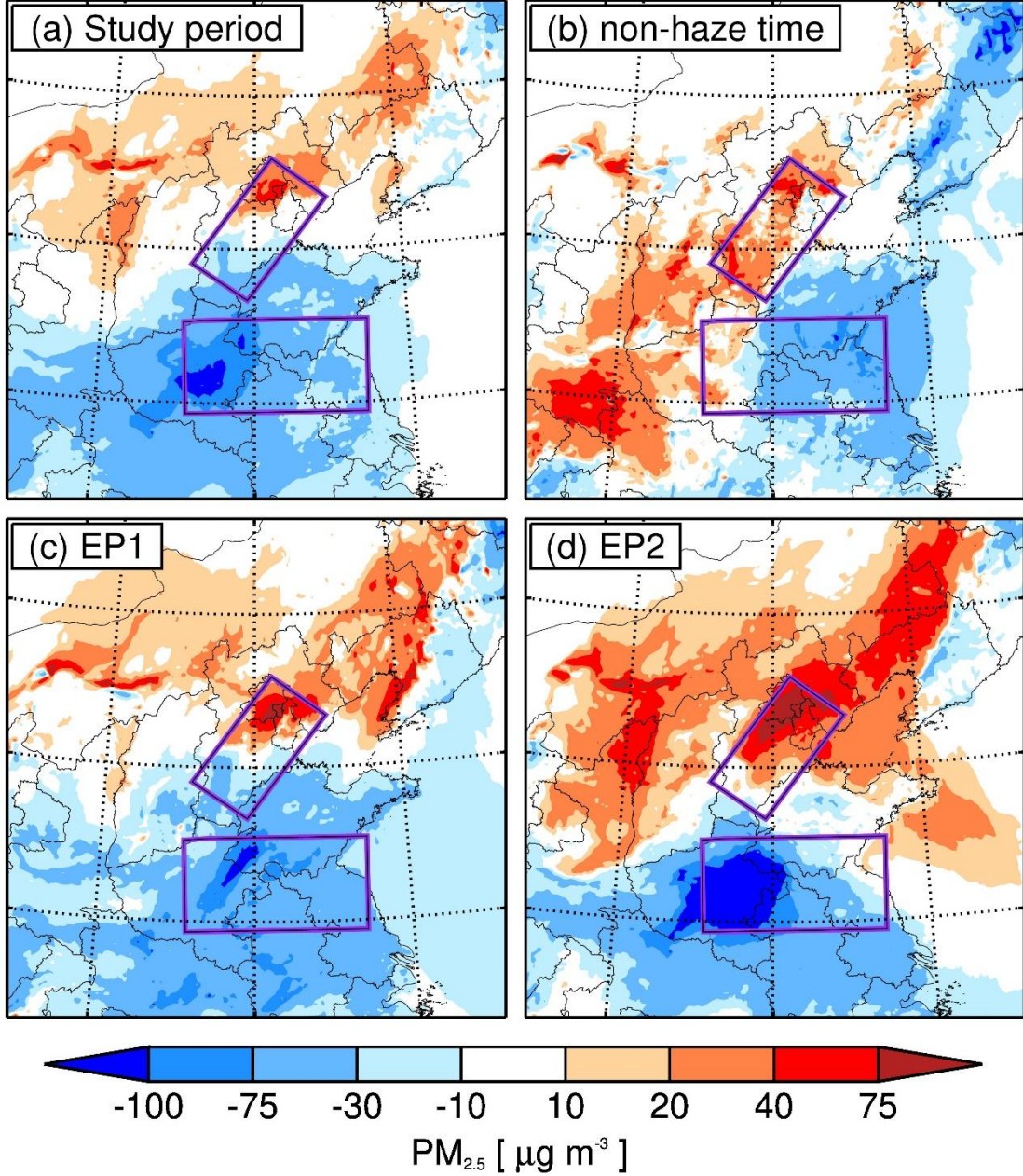

**Figure 7.** The pattern comparisons of the "BASE" simulation minus the "EMIS_METEO" simulation. The color gradient
represents coupled effects on PM$_{2.5}$ averaged from (a) the entire study period, (b) the non-haze period, (c) the EP1 haze
period, and (d) the EP2 haze period.

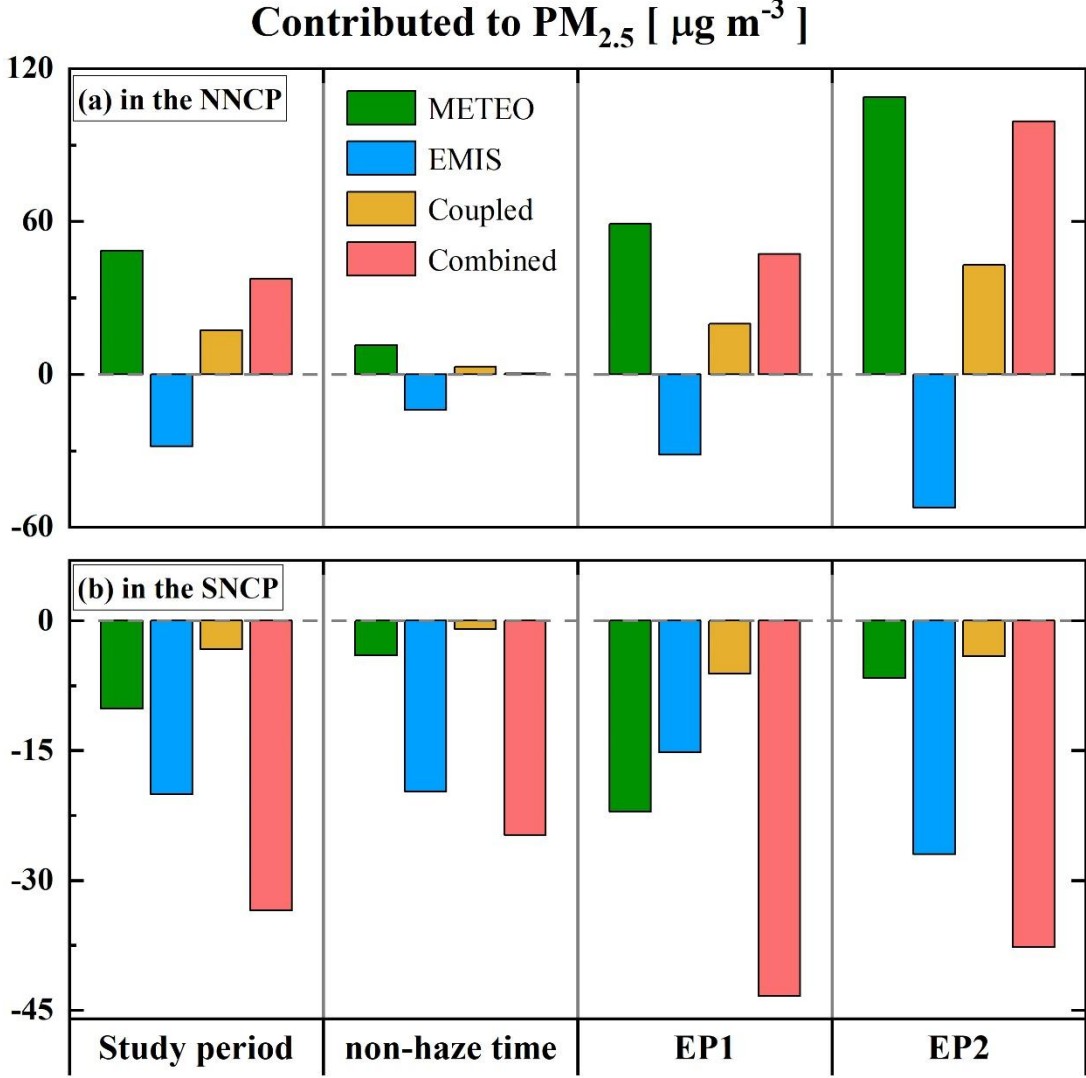

**Figure 8.** Regional contributions to PM$_{2.5}$ averaged in (a) the NNCP and (b) the SNCP during the entire period, non-haze
period, EP1, and EP2. The contributions include meteorological conditions (METEO), abrupt anthropogenic emissions
(EMIS) decreases, and coupled and combined effects of METEO and EMIS.
**Table 1**
**Table 1** Configurations of simulation cases in this study

| Experiments | Emission inventory | Meteorological field |
|:---:|:---:|:---:|
| BASE | 2020 | 2020 |
| SNCP0 | 2020, but with SNCP emissions set to zero | 2020 |
| METEO | 2020 | Mean over 2015 to 2019 |
| EMIS | 2019 | 2020 |
| EMIS_METEO | 2019 | Mean over 2015 to 2019 |



**Table 2**
**Table 2.** The statistical parameters of model performance include temporal assessments of *MB*, and *IOA* in the NNCP and
SCNP and at the IAP monitoring site.

| Statistical parameters | *NMB* | *IOA* |
|:---:|:---:|:---:|
| In the NNCP region | | |
| $PM_{2.5}$ | −5.6% | 0.91 |
| $SO_2$ | 4.8% | 0.82 |
| $O_3$ | 4.4% | 0.86 |
| $NO_2$ | 2.3% | 0.82 |
| CO | 1.5% | 0.85 |
| In the SNCP region | | |
| $PM_{2.5}$ | −2.1% | 0.86 |
| $SO_2$ | −11.0% | 0.76 |
| $O_3$ | −10.2% | 0.88 |
| $NO_2$ | 0.1% | 0.87 |
| CO | 6.0% | 0.79 |
| At the IAP monitoring site | | |
| Organic | 15.0% | 0.84 |
| Nitrate | −18.9% | 0.88 |
| Sulfate | −37.7% | 0.81 |
| Ammonium | −23.6% | 0.87 |
