# Peer review of "Impacts of meteorology and emission reductions on haze pollution"

_EGUsphere, 2024_

## Referee Comment (RC3)

**General comments**

This work analyzed the different responses of haze events over the northern and southern NCP during COVID lockdown. The analyses and interpretation were conducted through sensitivity tests of emissions and meteorological fields. It was demonstrated that pervasive emission reduction during COVID lockdown synergistically reduced the $PM_{2.5}$ pollution in the southern NCP, while it was counteracted by unfavorable meteorological conditions in the northern NCP leading to worse haze events. The methods are sound, and conclusions are important, while the interpretation needs to be strengthened. I recommend minor revisions before publishing at ACP.

**Specific comments**

1. It is argued in the title the work contains insights from six-year simulations, while the main text is heavily based on the analysis for the 3-week (Jan 21 to Feb 16 in the year of 2020) simulation. To avoid exaggerating insights, it is better to remove "Insights from six-year simulation" in the title.
2. There are two observational datasets used in the paper. The location of the IAP sites should be marked in the map as other sites shown in Figure 1.
3. Line 109, what are the temporal and spatial resolutions of the emission input?
4. For lines 110-114, it is argued that SNCP has significantly higher emissions than the NNCP, which is not evident from Figure S1. The spatial coverage of SNCP is larger than NNCP. The comparison should be done for region-averaged emission flux per square meter per second. Please show the direct statistical evident of higher emission flux in SNCP than NNCP.
5. For lines 115-118, this is not directly related to dataset description, but rather comments for the topographical characteristics. Consider removing it.
6. What is the spin-up time for the WRF-Chem simulation? Please clarify.
7. For lines 137-139, besides the initial and boundary meteorological conditions, are the meteorological fields within the spatial domain directly simulated by the WRF-Chem, or is it externally provided by NCEP FNL? If it is directly simulated by the WRF-Chem, please provide more details such as the advection, convection, and boundary layer mixing schemes as the effects of meteorological conditions are a main part of this paper. If it is externally provided by NCEP FNL, please add clarification.
8. For lines 146-157, how are the scaling factors determined for emission sensitivity test? Please provide rational for the determination of scaling factor. Is it determined relative to certain emissions? Consider adding demonstration that the emissions are back to normal after the scaling? How are the scaling factors applied for emission sensitivity test? Are they applied as a constant for each city?
9. Line 187-193, the low-biased sulfate concentrations were attributed to incomplete $SO_2$ oxidation pathway in the WRF-Chem in the paper. But the author showed that $SO_2$

shows great agreements against observations in Figure S3d with NMB 4.8%. If the sulfate underestimation were due to incomplete $SO_2$ oxidation, underestimation of $SO_2$ would be introduced. Please explain.

10. For section 3.1, are the simulated concentrations sampled at each site, or each city and then averaged to get regional mean, or directly simulation average for each region? Please clarify.

11. It is better to show the corresponding scatter plot for each region of NNCP and SNCP as that for all sites in Figure 2.

12. Figure 2 caption indicates simulated wind fields which are not shown in Figure 2.

13. Line 228, please add the definition of haze events. Is the criterion of 100 $\mu g/m^3$ $PM_{2.5}$ used?

14. Line 268, the statement of $PM_{2.5}$ levels of -50 $\mu g/m^3$ is confusing as concentration will never be negative. Consider clarifying that it is the effects of meteorological fields on the $PM_{2.5}$ concentration difference.

15. Line 292-294, the statement of regional transport of $PM_{2.5}$ from SNCP to NNCP does not have strong evidence. There is no prior $PM_{2.5}$ pollution outbreak in advance in SNCP showed in Figure 3 before EP2 pollution in NNCP. Northward winds are not necessarily indicating pollution transport from SNCP to NNCP when SNCP is clean. Direct evidence may be needed by conducting a sensitivity test by eliminating SNCP emissions and evaluate the $PM_{2.5}$ differences from that with SNCP emissions.

16. Line 301-308, please add more specific evidence of how the increased T2 improves which chemical reaction rates and how higher RH promote particle formation? Is there any direct evidence in this study?

17. Line 319-321, is there any direct evidence that for COVID lockdown period in this study it is also true that it is in a $NO_x$-saturated regime with reduced $HO_x$ concentrations? Please add direct evidence in this study.

18. Line 326-327, if the prior argument that NNCP is in a $NO_x$-saturated regime is true, then reduction of $NO_x$ does not necessarily lead to a change of $O_3$ concentration.

**Technical corrections**

1. Line 245, replace the bell symbol by bell-shaped.

---

## Author Comment (AC1)

**Response to Reviewer #3**

*General Comments*

*This work analyzed the different responses of haze events over the northern and southern NCP during COVID lockdown. The analyses and interpretation were conducted through sensitivity tests of emissions and meteorological fields. It was demonstrated that pervasive emission reduction during COVID lockdown synergistically reduced the PM2.5 pollution in the southern NCP, while it was counteracted by unfavorable meteorological conditions in the northern NCP leading to worse haze events. The methods are sound, and conclusions are important, while the interpretation needs to be strengthened. I recommend minor revisions before publishing at ACP.*

Thank you for the positive feedback and constructive suggestions. We appreciate the recognition of the soundness of our methods and the importance of our conclusions. In response, we strengthened the interpretation of our results, enhancing the clarity and depth of the discussion in the manuscript.

We respond to each specific comment in detail below. The reviewers' comments are shown in *black italics*. Our replies are in indented black text, and the modified text is in blue. The annotated line numbers refer to the revised copy of the manuscript.
* * *
*Specific comments:*

*Specific.1    It is argued in the title the work contains insights from six-year simulations, while the main text is heavily based on the analysis for the 3-week (January 21y 21 to February 16ry 16 in the year of 2020) simulation. To avoid exaggerating insights, it is better to remove "Insights from six-year simulation" in the title.*

Thank you for your constructive feedback. The phrase "insights from six-year simulations" in the original title was intended to highlight the climatological averages from 2015 to 2019, which provide a critical baseline for understanding the $PM_{2.5}$ dynamics during the one-month COVID-19 lockdown period. To address this and ensure clarity, we have revised the title and added detailed explanations throughout the manuscript.

**[*Title*]:**

"Impacts of meteorology and emission reductions on haze pollution during the Lockdown in the North China Plain"

**[Lines 189 in *Sect 2.2*]:**

In the METEO case, we applied the same emission inventory as the BASE case but with averaged meteorological conditions from 2015 to 2019. These mean meteorological fields were derived by averaging key meteorological variables (**Text S2**).

**[*Text S2*]:**

**Text S2 Mean meteorology from 2015 to 2019**

This study's mean meteorology field data was derived by averaging key meteorological variables (e.g., temperature, wind speed, relative humidity, and pressure) from 2015 to 2019. Given that the vertical levels in the NCEP FNL data varied across different years, we did not average the original data directly. Instead, we processed the data using the WRF Preprocessing System (WPS) to ensure consistency. Specifically, we ran WPS yearly to generate the met_em* files containing processed meteorological variables at uniform vertical levels and grid resolution. We then averaged these met_em* files across the six years at each grid point and pressure level, which helped preserve the atmospheric variables' vertical structure and physical coherence. This approach maintained a realistic representation of the atmospheric state by accounting for the multi-year variability while ensuring that the averaged fields were consistent with the WRF-Chem grid resolution. As the WPS processing already matched the data to the model's spatial resolution, no additional interpolation was required, thus ensuring the physical and spatial consistency of the averaged climatological fields used in the WRF-Chem simulations. This multi-year climatological averaging was designed to capture the typical variations in initial and boundary meteorological conditions. This approach provided a robust and representative baseline for multiple years, effectively minimizing the influence of anomalies or extreme weather events characteristic of any individual year.
* * *
*Specific.2 There are two observational datasets used in the paper. The location of the IAP sites should be marked in the map as other sites shown in Figure 1.*

Thank you for your helpful suggestion. We revised Figure 1 to include the location of the IAP monitoring site.

**[*Figure 1*]:**

[Figure]

Figure 1. The simulation domain in WRF-Chem, including topography. Circles represent the locations of cities with ambient air quality monitoring sites, with circle size reflecting the number of monitoring sites per city. The IAP observation sites are marked with black pentagons. The regions of interest, NNCP (Northern North China Plain) and SNCP (Southern North China Plain), are highlighted.

**[Line 115 in *Sect. 2.1*]:**

The second dataset includes chemical compositions such as organic matter, nitrate, sulfate, and ammonium, collected at the Institute of Atmospheric Physics (IAP), Chinese Academy of Sciences in Beijing, China (39°58′28″ N, 116°22′16″ E).
* * *
***Specific.3*** *Line 109, what are the temporal and spatial resolutions of the emission input?*

We added the specific temporal and spatial resolutions of the emission data. Thank you.

**[Lines 119 in *Sect. 2.1*]:**

We used the Multi-resolution Emission Inventory for China (MEIC), developed by Tsinghua University, with 2016 as the base year (http://meicmodel.org). This emission inventory includes emissions from power plants, transportation, industry, agriculture, and residential activities, with data available at a monthly time scale and a spatial resolution of 6 km. We updated the MEIC inventory to reflect the total provincial emissions estimated for 2020, using near-real-time estimation (Zheng et al., 2021). While the total emissions for each province were updated, the spatial distribution of emissions within each province still followed the intensity proportions from the 2016 MEIC inventory.
* * *
***Specific.4*** *For lines 110-114, it is argued that SNCP has significantly higher emissions than the NNCP, which is not evident from Figure S1. The spatial coverage of SNCP is larger than NNCP. The comparison should be done for region-averaged emission flux per square meter per second. Please show the direct statistical evident of higher emission flux in SNCP than NNCP.*

Thank you for your comment, and we apologize for the confusion caused by the unclear wording. Our intention was not to compare the emissions between SNCP and NNCP directly. Instead, we aimed to highlight that both regions exhibit significantly higher emission levels when compared to the northwestern part of the study area. We have revised the text to clarify this distinction and avoid misinterpretation.

**[Lines 132 in *Sect. 2.1*]:**

The spatial distribution of primary particles ($PM_{2.5}$) and gaseous pollutants ($CO$, $SO_2$, $NO_x$, $NH_3$, and $HCHO$) reveals significantly elevated emission levels across both the NNCP and the SNCP, particularly when compared to the less industrialized northwestern regions of the study area (**Figure S1**). These elevated emissions are primarily driven by dense urbanization and significant industrial activity (Zheng et al., 2021). The topographical features of the NCP, with higher elevations in the north and lower elevations in the south (**Figure 1**), along with substantial pollutant emissions from southern regions, indicate that under persistent southerly winds, pollutants are efficiently transported northward. This northward movement exacerbates air quality degradation, contributing to severe haze episodes in

the NNCP, intensifying regional air quality challenges, and complicating mitigation efforts(Huang et al., 2021).
* * *
***Specific.5*** *For lines 115-118, this is not directly related to dataset description, but rather comments for the topographical characteristics. Consider removing it.*

Removed as suggested. Thank you.
* * *
***Specific.6*** *What is the spin-up time for the WRF-Chem simulation? Please clarify.*

We added the spin-up time to the manuscript. Thank you

**[Lines 175 in *Sect. 2.2*]:**

For the episode simulations, the spin-up time is 3 days.
* * *
***Specific.7*** *For lines 137-139, besides the initial and boundary meteorological conditions, are the meteorological fields within the spatial domain directly simulated by the WRF-Chem, or is it externally provided by NCEP FNL? If it is directly simulated by the WRF-Chem, please provide more details such as the advection, convection, and boundary layer mixing schemes as the effects of meteorological conditions are a main part of this paper. If it is externally provided by NCEP FNL, please add clarification.*

Thank you for your valuable question. The meteorological fields within the spatial domain are directly simulated by the WRF-Chem model, rather than being externally provided by NCEP FNL. We have added the requested details about the advection, convection, and boundary layer mixing schemes.

**[Lines 151 in *Section 2.2*]:**

Further details regarding the model settings, initial and lateral meteorological and chemical fields, and anthropogenic and biogenic emission inventory(**Table S1**). We used physical schemes of the WRF single-moment(WSM) 6-class graupel microphysical scheme(Hong and Lim, 2006), the Mellor–Yamada–Janjic (MYJ) turbulent kinetic energy planetary boundary layer scheme (Janić, 2001), the unified Noah land-surface model (Chen and Dudhia, 2001) and the Monin-Obukhov surface layer scheme (Janić, 2001).

**[*Table S1*]:**

**Table S1** Model configuration for the simulation domain, meteorological schemes, chemical mechanisms, initial and lateral conditions, and emission inventories.

| | |
|---|---|
| **Domain** | |
| Size | 300 × 300 horizontal grid cells |
| Center | 116°E, 38° N |
| Horizontal resolution | 6 km × 6 km |
| Vertical resolution | 35 vertical levels, uneven intervals, spacing ranging from ~50 m near the surface, ~500 m at 2.5 km above the ground level, and more than 1 km at 14 km above the ground level |
| **Meteorology** | |
| Microphysics scheme | WSM 6-class graupel microphysics scheme (Hong and Lim, 2006) |
| Boundary layer scheme | MYJ PBL scheme (Janjić, 2002) |
| Surface layer scheme | Monin-Obukhov surface layer scheme (Janjić, 2002) |
| Land-surface scheme | Noah land-surface model (Chen and Dudhia, 2001) |
| Longwave radiation scheme | Goddard (Dudhia, 1989) |
| Shortwave radiation scheme | Goddard (Dudhia, 1989) |
| Dry deposition | Wesely (1989) |
| Wet deposition | CMAQ (Binkowski and Roselle, 2003) |
| **Chemistry** | |
| Gas phase chemistry | SAPRC99 chemical mechanism (Binkowski and Roselle, 2003) |
| Inorganic aerosols | ISORROPIA version 1.7 (Nenes et al., 1998) |
| Secondary organic aerosol | Nontraditional VBS parametrization (Li et al., 2011) |
| Photolysis rates | FTUV radiation transfer model (Tie et al., 2003) |
| **Boundary and initial conditions** | |
| Meteorological | NCEP FNL 6-hr 1° × 1° analysis data |
| Chemical | CAM-chem 6-hr outputs |
| **Emission inventory** | |
| Anthropogenic | MEIC (Zhang et al. 2009; Li et al., 2017) |
| Biogenic | MEGAN (Guenther et al., 2006) |
* * *
***Specific.8*** *For lines 146-157, how are the scaling factors determined for emission sensitivity test? Please provide rational for the determination of scaling factor. Is it determined relative to certain emissions? Consider adding demonstration that the emissions are back to normal after the scaling? How are the scaling factors applied for emission sensitivity test? Are they applied as a constant for each city?*

Thank you for pointing out the need for further clarification regarding the determination and application of scaling factors in the emission sensitivity test. We have revised the manuscript to provide a more detailed explanation of the emission inventory of the BASE and EMIS cases.

**[Lines 119 in *Sect. 2.1*]:**

We used the Multi-resolution Emission Inventory for China (MEIC), developed by Tsinghua University, with 2016 as the base year (http://meicmodel.org). This emission inventory includes emissions from power plants, transportation, industry, agriculture, and residential activities, with data available at a monthly time scale and a spatial resolution of 6 km. We updated the MEIC inventory to reflect the total provincial emissions estimated for 2020, using near-real-time estimation (Zheng et al., 2021). While the total emissions for each province were updated, the spatial distribution of emissions within each province still followed the intensity proportions from the 2016 MEIC inventory. Subsequently, we applied a top-down approach to adjust further the emission inventory, iteratively comparing model simulations with observed data to refine the estimates until the simulations closely matched the observations. We validated the final emission inventory using statistical parameters, including normalized mean bias (*NMB*), index of agreement (*IOA*), and correlation coefficient (*r*) (**Text S1**).

**[Lines 185 in *Section 2.2*]:**

In the EMIS experiment, we used the anthropogenic emission inventory from the BASE case. Still, we excluded any abrupt decreases associated with anthropogenic emission reductions during the COVID-19 lockdown period in 2020, following the provincial emission reduction ratios provided by Huang et al. (2021) (**Table S2**).

**[Table S2]:**

**Table S2** Provincial emission reduction ratios during the COVID-19 lockdown period 2020 in the study area.

| Species / Province | CO | NO$_x$ | SO$_2$ | VOCs | PM$_{2.5}$ | BC | OC |
|---|---|---|---|---|---|---|---|
| Beijing | 22% | 45% | 26% | 45% | 18% | 46% | 8% |
| Tianjin | 21% | 38% | 20% | 41% | 14% | 22% | 6% |
| Hebei | 15% | 45% | 16% | 36% | 12% | 17% | 5% |
| Anhui | 14% | 56% | 22% | 31% | 11% | 22% | 4% |
| Inner Mongolia | 14% | 29% | 15% | 34% | 13% | 16% | 6% |
| Shaanxi | 19% | 45% | 18% | 34% | 13% | 22% | 5% |
| Hubei | 19% | 55% | 23% | 35% | 16% | 23% | 10% |
| Jilin | 16% | 39% | 23% | 34% | 13% | 18% | 5% |
| Liaoning | 21% | 40% | 28% | 36% | 16% | 28% | 8% |
| Henan | 23% | 57% | 22% | 41% | 18% | 35% | 8% |
| Shandong | 23% | 50% | 25% | 39% | 19% | 35% | 9% |
| Jiangsu | 23% | 50% | 26% | 41% | 16% | 35% | 7% |
| Shanghai | 35% | 48% | 42% | 45% | 34% | 54% | 42% |
* * *
***Specific.9*** *Line 187-193, the low-biased sulfate concentrations were attributed to incomplete SO2 oxidation pathway in the WRF-Chem in the paper. But the author showed that SO2 shows great agreements against observations in Figure S3d with NMB 4.8%. If the sulfate underestimation were due to incomplete SO2 oxidation, underestimation of SO2 would be introduced. Please explain.*

Thank you for this insightful comment. While the overall agreement for SO$_2$ across 65 stations in the NNCP is indeed high (NMB = 4.8%), the nearest monitoring site to the IAP location (within approximately 10 km) shows an SO$_2$ underestimation with an NMB of -12.1% (**Figure S4**). This underestimation pattern near the IAP site aligns with the low bias observed in sulfate concentrations at this location. These results suggest that while WRF-Chem captures regional SO$_2$ concentrations effectively, it may not adequately represent key localized oxidation processes, such as aqueous-phase reactions and heterogeneous transformations, which are crucial for sulfate formation, particularly in urban areas with high emission densities. This could explain why sulfate concentrations at specific locations are underpredicted, even though regional SO$_2$ levels strongly agree with observations (Liu et al., 2021; Song et al., 2019).

**[Lines 221 in *Sect. 3.1*]:**

On a regional scale, the model's good performance in $SO_2$ simulation (NMB = 4.8% in the NNCP) does not entirely explain the sulfate underprediction, particularly near the IAP site, where local $SO_2$ is underestimated by −12.1% (**Figure S4**). This local discrepancy suggests that WRF-Chem may inadequately capture oxidation processes such as aqueous-phase and metal-catalyzed reactions, leading to sulfate underestimation in urban areas with high pollution levels (Guo et al., 2017; Liu et al., 2017; Zheng et al., 2015). While the model effectively reproduces the temporal variability of critical components, the consistent underestimation of sulfate and ammonium indicates the need for further refinements in the representation of $SO_2$ emissions and associated oxidation pathways(Cheng et al., 2016; Li et al., 2018).

**[*References*]**

Guo, H., Liu, J., Froyd, K. D., Roberts, J. M., Veres, P. R., Hayes, P. L., Jimenez, J. L., Nenes, A., and Weber, R. J.: Fine particle pH and gas–particle phase partitioning of inorganic species in Pasadena, California, during the 2010 CalNex campaign, Atmospheric Chem. Phys., 17, 5703–5719, 2017.

Liu, M., Song, Y., Zhou, T., Xu, Z., Yan, C., Zheng, M., Wu, Z., Hu, M., Wu, Y., and Zhu, T.: Fine particle pH during severe haze episodes in northern China, Geophys. Res. Lett., 44, 5213–5221, 2017.

Zheng, G. J., Duan, F. K., Su, H., Ma, Y. L., Cheng, Y., Zheng, B., Zhang, Q., Huang, T., Kimoto, T., Chang, D., Pöschl, U., Cheng, Y. F., and He, K. B.: Exploring the severe winter haze in Beijing: the impact of synoptic weather, regional transport and heterogeneous reactions, Atmospheric Chem. Phys., 15, 2969–2983.

Cheng, Y., Zheng, G., Wei, C., Mu, Q., Zheng, B., Wang, Z., Gao, M., Zhang, Q., He, K., & Carmichael, G. (2016). Reactive nitrogen chemistry in aerosol water as a source of sulfate during haze events in China. Science Advances, 2(12), e1601530.

Li, L., Hoffmann, M. R., and Colussi, A. J.: Role of nitrogen dioxide in the production of sulfate during Chinese haze-aerosol episodes, Environ. Sci. Technol., 52, 2686–2693, 2018.

[Figure]

**$SO_2$ concentration near IAP site [ μg m$^{-3}$ ]**

[Figure]

**Figure S4**. Comparisons of simulated and observed mass concentrations of $SO_2$ near the IAP site monitoring site from January 21 to February 16, 2020. Blue lines represent simulated concentrations, while red lines indicate observed concentrations.
* * *
*Specific.10 For section 3.1, are the simulated concentrations sampled at each site, or each city and then averaged to get regional mean, or directly simulation average for each region? Please clarify.*

Thank you for your question. We clarified that the simulated concentrations were first sampled at each regional observational site.

**[Lines 129 in *Sect. 2.2*]:**

The simulated concentrations were first sampled at each observational site within the region. These site-specific concentrations were then averaged to calculate the regional mean for the NNCP and SNCP, respectively.
* * *
*Specific.11 It is better to show the corresponding scatter plot for each region of NNCP and SNCP as that for all sites in Figure 2.*

Thank you for your suggestion. We have added the corresponding scatter plots for NNCP and SNCP in **Figure S4**, which show statistical comparisons for key pollutants.

**[Lines 240 in *Sect. 3.1*]:**

The spatial distributions of simulated and observed gaseous pollutants, averaged over the episode, demonstrated strong spatial consistency, with correlation coefficients (*r*) of 0.67 for $O_3$, 0.86 for $SO_2$, and 0.77 for $NO_2$ across the research domain (**Figure 2e, 2f**). This high consistency was also observed in the NNCP and SNCP regions (**Figure S5**), with correlation coefficients for $PM_{2.5}$ and $O_3$ of 0.98 and 0.71 in the NNCP, and 0.94 and 0.67 in the SNCP. Similarly, the correlation coefficients for $SO_2$ and $NO_2$ were 0.77 and 0.83 in the NNCP, and 0.89 and 0.82 in the SNCP.

[Figure]

**Figure S5.** Statistical comparisons of model simulations and observations for (a) PM$_{2.5}$ and O$_3$, and (b) SO$_2$ and NO$_2$ in the NNCP and SNCP regions.

\-\-\-\-\-\-\-\-\-\-\-\-\-\-\-\-\-\-\-\-\-\-\-\-\-\-\-\-\-\-\-\-\-\-\-\-\-\-\-\-\-\-\-\-\-\-\-\-\-\-\-\-\-\-\-\-\-\-\-\-\-\-\-\-\-\-\-\-\-\-\-\-\-\-\-\-\-\-\-\-\-\-\-\-\-\-\-\-\-\-\-

*Specific.12 Figure 2 caption indicates simulated wind fields which are not shown in Figure 2.*

We revised the caption for Figure 2. Thank you.

\-\-\-\-\-\-\-\-\-\-\-\-\-\-\-\-\-\-\-\-\-\-\-\-\-\-\-\-\-\-\-\-\-\-\-\-\-\-\-\-\-\-\-\-\-\-\-\-\-\-\-\-\-\-\-\-\-\-\-\-\-\-\-\-\-\-\-\-\-\-\-\-\-\-\-\-\-\-\-\-\-\-\-\-\-\-\-\-\-\-\-

*Specific.13 Line 228, please add the definition of haze events. Is the criterion of 100 µg/m3 PM2.5 used?*

Thank you for the comment. We included the specific dates for each episode in the manuscript.

**[Lines 261 in *Sect. 3.2*]:**

Despite the significant reduction in anthropogenic emissions and lower concentrations of NO$_2$ and SO$_2$, two unexpected heavy haze episodes occurred in the NNCP. Here, we defined haze events as periods when the daily average PM$_{2.5}$ concentration in the NNCP exceeds 100 µg m$^{-3}$.

\-\-\-\-\-\-\-\-\-\-\-\-\-\-\-\-\-\-\-\-\-\-\-\-\-\-\-\-\-\-\-\-\-\-\-\-\-\-\-\-\-\-\-\-\-\-\-\-\-\-\-\-\-\-\-\-\-\-\-\-\-\-\-\-\-\-\-\-\-\-\-\-\-\-\-\-\-\-\-\-\-\-\-\-\-\-\-\-\-\-\-

*Specific.14 Line 268, the statement of PM2.5 levels of -50 µg/m3 is confusing as concentration will never be negative. Consider clarifying that it is the effects of meteorological fields on the PM2.5 concentration difference.*

Thank you for the suggestion. The negative value refers to the decrease in PM$_{2.5}$ concentrations due to the influence of meteorological fields. We clarified this in the text.

Meteorological factors significantly influenced PM$_{2.5}$ concentrations during the study period, as illustrated by the pattern comparisons between the "BASE" and "METEO" simulations (**Figure 5a**). Changes in PM$_{2.5}$ concentrations ranged from decreases of up to 50 µg m$^{-3}$ to increases exceeding 100 µg m$^{-3}$, revealing an apparent north-south disparity.
* * *
***Specific.15*** *Line 292-294, the statement of regional transport of PM2.5 from SNCP to NNCP does not have strong evidence. There is no prior PM2.5 pollution outbreak in advance in SNCP showed in Figure 3 before EP2 pollution in NNCP. Northward winds are not necessarily indicating pollution transport from SNCP to NNCP when SNCP is clean. Direct evidence may be needed by conducting a sensitivity test by eliminating SNCP emissions and evaluate the PM2.5 differences from that with SNCP emissions.*

Thank you for your insightful suggestion. To provide clearer evidence of the regional transport of PM$_{2.5}$ from SNCP to NNCP, we conducted the "SNCP0" simulation by setting SNCP emissions to zero in the BASE scenario. We quantitatively showed the difference in PM$_{2.5}$ concentrations between the "SNCP0" and "BASE" simulations across different periods, highlighting the contribution of SNCP emissions to PM$_{2.5}$ levels in NNCP.

To quantify the influence of SNCP emissions on PM$_{2.5}$ concentrations in NNCP, we also performed an additional sensitivity test (SNCP0) by setting SNCP emissions to zero within the BASE scenario.

Meanwhile, the comparison between the "SNCP0" simulation (with SNCP emissions set to zero) and the "BASE" case demonstrated a substantial reduction in PM$_{2.5}$ concentrations in the NNCP (**Figure S8**), particularly during EP2. This reduction, ranging from 15 to 30 µg m$^{-3}$ in certain areas of the NNCP (**Figure S8b**), provides direct evidence that SNCP emissions contribute significantly to PM$_{2.5}$

accumulation in the NNCP via northward transport. This finding underscores the importance of regional transport, facilitated by northward winds, in elevating PM$_{2.5}$ concentrations in the NNCP, especially under meteorological conditions that support pollutant movement from south to north.

[Figure]

**Figure S7.** The pattern comparisons of "SNCP0" simulations minus the "BASE" simulation. The color gradient represents PM$_{2.5}$ changes averaged from (a) the EP1 haze period, and (b) the EP2 haze period,
* * *
*Specific.16* *Line 301-308, please add more specific evidence of how the increased T2 improves which chemical reaction rates and how higher RH promote particle formation? Is there any direct evidence in this study?*

Thank you for this insightful comment. We have clarified in the revised text that the SEN_METEO simulation captures the influence of elevated temperature (T2) and relative humidity (RH) on secondary aerosol formation through the online WRF-Chem model. Specifically, increased T2 enhances gas-phase oxidation reactions, contributing to secondary organic aerosol (SOA) and nitrate formation, while elevated RH fosters aqueous-phase chemistry that promotes sulfate formation on particle surfaces. Our online WRF-Chem model integrates the effects of T2 and RH directly into the PM$_{2.5}$ concentration results, providing evidence that meteorological factors remained influential in sustaining elevated PM$_{2.5}$ levels during haze episodes, despite

reduced emissions. This highlights the importance of considering meteorological conditions alongside emission reductions in air quality management.

**[Lines 345 in *Sect. 3.3*]:**

Regional variations in haze episodes underscore the critical role of elevated near-surface temperature (T2) and relative humidity (RH) in driving secondary aerosol formation (**Figure S9**). In the NNCP, elevated T2 accelerates gas-phase oxidation reactions, converting volatile organic compounds (VOCs) and nitrogen oxides ($NO_x$) into secondary organic aerosols (SOAs) and nitrate aerosols, thus contributing to increased $PM_{2.5}$ levels despite reduced emissions (Huang et al., 2021; Seinfeld and Pandis, 2016). Similarly, elevated RH facilitates aqueous-phase reactions that convert $SO_2$ into sulfate on particle surfaces, aided by aerosol liquid water, and this effect is particularly pronounced during haze episodes, where high RH accelerates sulfate formation even with decreased emissions (Le et al., 2020; Wang et al., 2020). The online WRF-Chem model captures these interactions in the SEN_METEO simulation, integrating the effects of T2 and RH into the modeled $PM_{2.5}$ concentrations. Although the study does not isolate each specific chemical pathway, the correlation between elevated T2, RH, and higher $PM_{2.5}$ concentrations aligns with previous research, and underscores the pivotal role of meteorological conditions in secondary aerosol formation. This finding highlights the importance of considering meteorological influences in addition to emission reductions, as unfavorable weather conditions can offset the expected improvements from reduced emissions and sustain elevated $PM_{2.5}$ levels. This understanding is essential for developing effective air pollution control strategies that account for emissions and meteorological variability.

**[*References*]:**

Huang, X., Ding, A., Gao, J., Zheng, B., Zhou, D., Qi, X., ... & He, K. (2021). Enhanced secondary pollution offset reduction of primary emissions during COVID-19 lockdown in China. National Science Review, 8(1), nwaa137.

Seinfeld, J. H., & Pandis, S. N. (2016). Atmospheric Chemistry and Physics: From Air Pollution to Climate Change. John Wiley & Sons.

Le, T., Wang, Y., Liu, L., Yang, J., Yung, Y. L., Li, G., & Seinfeld, J. H. (2020). Unexpected air pollution with marked emission reductions during the COVID-19 outbreak in China. Science, 369(6504), 702-706.

Wang, Y., Yuan, Y., Wang, Q., Liu, C., Zhi, Q., & Cao, J. (2020). Changes in air quality related to the control of coronavirus in China: Implications for traffic and industrial emissions. Science of the Total Environment, 731, 139133.
* * *
***Specific.17*** *Line 319-321, is there any direct evidence that for COVID lockdown period in this study it is also true that it is in a NOx-saturated regime with reduced HOx concentrations? Please add direct evidence in this study.*

Thank you for your comment. We clarified that wintertime ozone production in northern China's urban areas generally occurs in a NOx-saturated regime due to limited HOx radicals and low solar radiation (Seinfeld & Pandis, 2016). During the COVID-19 lockdown, significant NOx reductions reduced ozone titration, allowing ozone concentrations to reach about 65.7 µg/m³, even when PM₂.₅ exceeded 100 µg/m³ (Figure S12). This aligns with findings that reduced NOx can lead to increased ozone levels in NOx-saturated environments, with additional influences from aerosol radiative effects and precursor interactions (Levy et al., 2014; Wu et al., 2020; Le et al., 2020). This supports our conclusion that the NNCP remained NOx-saturated during the lockdown.

**[Lines 381 in *Sect 3.4*]:**

Wintertime ozone production in urban areas of northern China typically occurs in a $NO_x$-saturated regime, primarily due to a lack of HOx radicals and limited solar radiation during winter(Seinfeld and Pandis, 2016). Additionally, reduced fresh NO emissions alleviate ozone titration(Levy et al., 2014). Thus, a reduction in $NO_x$ often leads to increased ozone levels. In the NCP during winter, there is usually an inverse relationship between $PM_{2.5}$ and $O_3$, attributed to the aerosol radiative effect on ozone photochemistry(Li et al., 2017b; Wu et al., 2020). However, during the COVID-19 lockdown, this inverse relationship disappeared in the NNCP, with ozone concentrations reaching approximately 65.7 µg m⁻³ even when $PM_{2.5}$ levels exceeded 100 µg m⁻³ (**Figure S12**). Significant reductions in $NO_x$ emissions reduced ozone titration, resulting in elevated ozone levels despite higher $PM_{2.5}$ concentrations. This pattern aligns with previous findings that in $NO_x$-saturated environments, reductions in $NO_x$ can increase ozone levels, with additional effects from aerosol radiative influences and precursor interactions shaping the $O_3-PM_{2.5}$ relationship(Le et al., 2020). These dynamics highlight the importance of considering nonlinear chemical and meteorological factors when assessing air quality responses to emission reductions.

**[*References*]:**

Seinfeld J H, Pandis S N. Atmospheric chemistry and physics: from air pollution to climate change[M]. John Wiley & Sons, 2016.

Levy M, Zhang R, Zheng J, et al. Measurements of nitrous acid (HONO) using ion drift-chemical ionization mass spectrometry during the 2009 SHARP field campaign[J]. Atmospheric environment, 2014, 94: 231-240.

Li G, Bei N, Cao J, et al. Widespread and persistent ozone pollution in eastern China during the non-winter season of 2015: observations and source attributions[J]. Atmospheric Chemistry and Physics, 2017, 17(4): 2759-2774.

Wu J, Bei N, Hu B, et al. Aerosol–photolysis interaction reduces particulate matter during wintertime haze events[J]. Proceedings of the National Academy of Sciences, 2020, 117(18): 9755-9761.

Le, T., Wang, Y., Liu, L., Yang, J., Yung, Y. L., Li, G., and Seinfeld, J. H.: Unexpected air pollution with marked emission reductions during the COVID-19 outbreak in China, Science, 369, 702–706, 2020.

[*Figure S12*]:

[Figure]

**Figure S12.** Daytime variation of $O_3$ and $NO_2$ (10:00 to 16:00 Beijing Time) as a function of $PM_{2.5}$ concentration during the study period in the NNCP.
* * *
***Specific.18*** *Line 326-327, if the prior argument that NNCP is in a NOx-saturated regime is true, then reduction of NOx does not necessarily lead to a change of O3 concentration.*

Thank you for the comment. In NOx-saturated regimes, reducing NOx emissions generally has a limited impact on $O_3$ production due to the prevailing chemical conditions where high NOx levels suppress ozone formation through titration. However, during the COVID-19 lockdown, NO emissions reduction alleviated this titration effect, allowing background ozone levels to rise even in a NOx-saturated environment. This dynamic is supported by observations where, despite NOx reductions, ozone concentrations increased, reaching approximately 65.7 $\mu g\ m^{-3}$ when $PM_{2.5}$ levels were high (**Figure S12**). This behavior aligns with previous

studies, which found that in NOx-saturated conditions, a decrease in NO emissions can lead to elevated ozone due to reduced titration (Seinfeld & Pandis, 2016; Le et al., 2020; Wu et al., 2020). This highlights the complex and nonlinear relationship between NOx and ozone, which is influenced by chemical and meteorological factors in urban northern China during winter.

**[Lines 381 in *Sect 3.4*]:**

Wintertime ozone production in urban areas of northern China typically occurs in a $NO_x$-saturated regime, primarily due to a lack of HOx radicals and limited solar radiation during winter(Seinfeld and Pandis, 2016). Additionally, reduced fresh NO emissions alleviate ozone titration(Levy et al., 2014). Thus, a reduction in $NO_x$ often leads to increased ozone levels. In the NCP during winter, there is usually an inverse relationship between $PM_{2.5}$ and $O_3$, attributed to the aerosol radiative effect on ozone photochemistry(Li et al., 2017b; Wu et al., 2020). However, during the COVID-19 lockdown, this inverse relationship disappeared in the NNCP, with ozone concentrations reaching approximately 65.7 $\mu g\ m^{-3}$ even when $PM_{2.5}$ levels exceeded 100 $\mu g\ m^{-3}$ (**Figure S12**). Significant reductions in $NO_x$ emissions reduced ozone titration, resulting in elevated ozone levels despite higher $PM_{2.5}$ concentrations. This pattern aligns with previous findings that in $NO_x$-saturated environments, reductions in $NO_x$ can increase ozone levels, with additional effects from aerosol radiative influences and precursor interactions shaping the $O_3-PM_{2.5}$ relationship(Le et al., 2020). These dynamics highlight the importance of considering nonlinear chemical and meteorological factors when assessing air quality responses to emission reductions.
* * *
***Technical corrections:***

***Technical.1*** *Line 245, replace the bell symbol by bell-shaped.*

Changed as suggested. Thank you.

---

## Author Comment (AC2)

**Response to Reviewer #1**

**General Comments**

*This manuscript presents a significant study examining the impacts of meteorology and emission reductions on PM2.5 levels during the COVID-19 lockdown in the North China Plain (NCP). The authors utilize the WRF-Chem model to investigate the complex interactions between anthropogenic emissions, meteorology, and air quality, revealing important regional disparities in PM2.5 responses between the Northern and Southern NCP. The analysis of how adverse meteorological conditions in the Northern NCP negated the benefits of emission reductions is particularly noteworthy. This manuscript aligns with the scope of ACP, and the methodology is sound. However, there are several areas that require enhancement, particularly in clarifying the research objectives, providing more detail in the methodology, and including the rationale for the selected model and specific parameters. I will recommend acceptance of the manuscript after the following minor concerns are addressed.*

> We appreciate the positive feedback and constructive suggestions. We have carefully reviewed the comments and taken steps to enhance the manuscript. In response, we have made the following revisions and clarifications: (1) Clarifying the research objectives; (2) Providing more detail in the methodology; and (3) Rationale for model selection and specific parameters. We respond to the concerns in detail below.

> We respond to each specific comment in detail below. The reviewers' comments are shown in *black italics*. Our replies are in indented black text, and the modified text is in blue. The annotated line numbers refer to the revised copy of the manuscript.
* * *
**Major comments:**

***Major.1*** *The relationship between air pollution and emission reduction during the COVID-19 lockdown in China is a notable case in air pollution control; however, several existing studies exist on this topic. It is recommended that the author further enhance the discussion by more robustly comparing the results of this study with prior research, thereby underscoring the distinctive contributions of this paper.*

> Thank you for your valuable feedback on the novelty of our study. We have revised the manuscript to highlight our approach's unique contributions, particularly the methodologies used and the regional insights provided. Specifically, we clarified how using the WRF-Chem model with sensitivity experiments (e.g., SEN_METEO_EMIS) and factor separation methods offers a new perspective on the

relationship between meteorology and emission reductions. Additionally, we emphasized our findings on the differing responses between the northern and southern North China Plain (NCP), which enhanced the understanding of air quality dynamics during the lockdown.

**[Lines 24 in *Abstract*]:**

Our analysis highlights a marked regional contrast: in the Northern NCP (NNCP), adverse meteorology largely offset emission reductions, resulting in $PM_{2.5}$ increases of 30 to 60 μg m$^{-3}$ during haze episodes. Conversely, the Southern NCP (SNCP) benefited from favourable meteorological conditions that lowered $PM_{2.5}$ by 20 to 40 μg m-3, combined with emission reductions. These findings emphasize the critical role of meteorology in shaping the air quality response to emission changes, particularly in regions like the NNCP, where unfavourable weather patterns can counteract the benefits of emission reductions. Our study provides valuable insights into the complex interplay of emissions, meteorology, and pollutant dynamics, suggesting that adequate air quality strategies must integrate emissions controls and meteorological considerations to address regional variations effectively.

**[Lines 79 in *Introduction*]:**

We emphasize the localized differences in how meteorological conditions and emission reductions affect air quality within the North China Plain, specifically between the Northern North China Plain (NNCP) and Southern North China Plain (SNCP). Utilizing the WRF-Chem model, we conducted detailed sensitivity experiments that allowed us to isolate and quantify the individual and combined impacts of emissions and meteorology on air quality, which can deepen the understanding of air quality dynamics in different regional contexts.

**[Lines 464 in *Conclusions*]:**

Previous studies have primarily focused on the overall impacts of meteorological conditions and emission reductions on air quality across the North China Plain and even nationwide. We emphasize the localized differences in how meteorological conditions and emission reductions affect air quality within the North China Plain, specifically between the NNCP and SNCP. Our findings underscore the critical role that meteorological conditions play in modulating the effects of emission reductions. The combination of unfavourable meteorological factors and emission reductions in the NNCP led to overall increases in $PM_{2.5}$ levels, with significant increases during haze episodes. Meanwhile, in the SNCP, meteorological conditions and emission reductions consistently contributed to lower $PM_{2.5}$ concentrations.
* * *
*Major.2 The authors have distinctly defined two regions of interest, namely the NNCP and the SNCP. Please elaborate on the spece two regions were designated as depicted*

*in Figure 1? What were the crucial factors that the authors took into account when defining the boundaries of the two regions?*

Thank you for the thoughtful comment. Strict geographical limits do not bind the delineation of the NNCP and SNCP; instead, it is based on representative features and differences critical for a comprehensive assessment of geographical, meteorological, and emission characteristics. The boundaries were drawn to effectively capture the distinct local attributes of each region, allowing for meaningful comparisons and insights into air quality dynamics. Regional differentiation is crucial for understanding the air quality dynamics across the NCP.

**[Lines 96 in *Section 2.1*]:**

We defined these regions by thoroughly analyzing geographical features, weather conditions, and emission sources. The NNCP, which generally includes the cities in the Beijing-Tianjin-Hebei (BTH) area, is surrounded by mountains and elevated terrain to the north and west. These features make it harder for pollutants to disperse, leading to pollutant buildup, especially in winter when stagnant atmospheric conditions dominate (Feng et al., 2020; Li et al., 2019). On the other hand, the SNCP is characterized by lower elevations and broad plains, which help disperse pollutants due to more vital wind patterns and higher planetary boundary layer heights (Huang et al., 2021). The emissions in these two regions also differ significantly. The NNCP is mainly affected by concentrated urban and industrial emissions from the BTH area. At the same time, the SNCP has a broader variety of sources, including industrial and agricultural emissions, creating a more diverse pollutant profile(Zheng et al., 2021). These differences in geography, weather, and emissions provide a basis for studying how meteorological factors and emission reductions affect air quality differently across the NCP (**Figure 1**). By examining these sub-regions separately, we can better understand how air quality interventions vary in effectiveness across different areas.

**[*References*]**

Feng, J., Liao, H., Li, Y., Zhang, Z., & Tang, Y. (2020). Long-term trends and variations in haze-related weather conditions in north China during 1980–2018 based on emission-weighted stagnation intensity. Atmospheric Environment, 240, 117830.

Li, J., Liao, H., Hu, J., & Li, N. (2019). Severe particulate pollution days in China during 2013–2018 and the associated typical weather patterns in Beijing-Tianjin-Hebei and the Yangtze River Delta regions. Environmental Pollution, 248, 74-81.

Huang, X., Ding, A., Gao, J., Zheng, B., Zhou, D., Qi, X., & He, K. (2021). Enhanced secondary pollution offset reduction of primary emissions during COVID-19 lockdown in China. National Science Review, 8(1), nwaa137.

Zheng, B., Zhang, Q., Geng, G., Chen, C., Shi, Q., Cui, M., ... & He, K. (2021). Changes in China's anthropogenic emissions and air quality during the COVID-19 pandemic in 2020. Earth System Science Data, 13(6), 2895-2907.
* * *
*Major.3 The authors mainly discuss the spatial differences in the impact of emissions and meteorology on the total PM2.5 concentrations, how about the chemical components within PM2.5, particularly secondary inorganic and organic aerosols? Do these chemical components exhibit the same spatial variation characteristics?*

Thank you for raising this critical point. We examined the spatial variations of chemical components within $PM_{2.5}$ on how the chemical components and added descriptions in the text.

**[Lines 362 in *Section 3.3*]:**

These meteorological effects also impact secondary aerosols, including secondary organic aerosols (SOAs) and secondary inorganic aerosols (SIAs), with substantial variability between the NNCP and SNCP regions. In the NNCP, stagnant conditions and reduced boundary layer heights limited pollutant dispersion, contributing to the accumulation of SOAs and SIAs. High humidity further exacerbated the formation of secondary aerosols, resulting in elevated concentrations (**Figure S10**). Conversely, the SNCP benefited from higher PBLH (**Figure S7**) and dynamic wind patterns(**Figure 4a**), which enhanced the dispersion of both primary and secondary aerosols, reducing their concentrations. Due to the very low emissions of biogenic secondary organic aerosol (BSOA) precursors during wintertime(Guenther et al., 2012), the BSOA contribution to $PM_{2.5}$ concentrations is insignificant, averaging less than 2 µg m$^{-3}$ throughout the study period (**Figure S11a**). The average BSOA accounted for less than 2% of total $PM_{2.5}$ mass in the BASE simulations (**Figure S11b**), indicating a minor role for biogenic emissions in shaping wintertime air quality.

**[Lines 377 in *Section 3.4*]:**

In addition to the overall $PM_{2.5}$ reductions, emission controls significantly impacted SOAs and SIAs in the NNCP and SNCP (**Figure S10b, 10d**). The reductions in SOAs and SIAs were driven by decreased availability of precursors such as VOCs for SOAs and $SO_2$ and $NO_x$ for SIAs(Huang et al., 2021).

**[Figure S10]**

[Figure]

**Figure S10.** Comparison of simulated changes in chemical components during the study period between the "BASE" scenario and two sensitivity cases: (a,c) " METEO" and (b,d) "EMIS". The chemical components include (a,c) secondary organic aerosols and (b,d) secondary inorganic aerosols (SIAs), including sulfate, nitrate, and ammonium.
* * *
*Minor comments:*

*Minor.1 Provide a rationale for using the WRF-Chem model, highlighting its advantages for simulating meteorological and chemical interactions. Include specific parameters used in the WRF-Chem model simulations, such as resolution, boundary conditions, and initial conditions. This detail will help readers understand the modeling approach and assess its performance*

Thank you for your constructive feedback. We provided a clear rationale for using the WRF-Chem model and detailing its setup parameters to facilitate a better understanding of our modeling approach and its strengths in capturing the complex interactions between meteorological processes and chemical transformations.

**[Lines 143 in *Section 2.2*]:**

We employed a specific version (version 3.5.1) of the WRF-Chem model (Grell et al., 2005). We chose the WRF-Chem model because it can simulate coupled atmospheric processes, including emissions, transport, chemical transformations, and aerosol-cloud interactions. This "online" approach allows for dynamic feedback between meteorological conditions and air pollutants. It is well-suited for assessing the interplay between emission reductions and meteorology on $PM_{2.5}$ concentrations during the COVID-19 lockdown period. The model's ability to simultaneously simulate meteorology and chemistry provides advantages over models that treat these processes separately, ensuring that interactions such as aerosol-radiation and aerosol-cloud effects are effectively captured (Li et al., 2011).

Further details regarding the model settings, initial and lateral meteorological and chemical fields, and anthropogenic and biogenic emission inventory(**Table S1**). We used physical schemes of the WRF single-moment(WSM) 6-class graupel microphysical scheme(Hong and Lim, 2006), the Mellor–Yamada–Janjic (MYJ) turbulent kinetic energy planetary boundary layer scheme (Janić, 2001), the unified Noah land-surface model (Chen and Dudhia, 2001) and the Monin-Obukhov surface layer scheme (Janić, 2001).

**[Lines 165 in *Section 2.2*]:**

The simulation domain, centered at (116 °E, 38 °N), consisted of 300 × 300 horizontal grid cells with a 6 km resolution (**Figure 1**). The vertical resolution consisted of 35 levels, extending from the surface to 50 hPa, allowing for a detailed representation of boundary layer processes and pollutant dispersion. The initial and boundary meteorological conditions were derived from the National Centers for Environmental Prediction (NCEP) Final (FNL) reanalysis data at a 1° × 1° spatial resolution and six-hour temporal intervals (Kalnay et al., 2018). Chemical initial and boundary conditions were interpolated from the CAM-Chem (Community Atmosphere Model with Chemistry) global chemistry model(Danabasoglu et al., 2020). The anthropogenic emissions inventory for 2020 was based on a bottom-up approach, incorporating near-real-time data (Zheng et al., 2021), and biogenic emissions were computed online using the Model of Emissions of Gases and Aerosols from Nature (MEGAN)(Guenther et al., 2006). For the episode simulations, the spin-up time is 3 days.

**[*References*]**

Hong, S.-Y., and Lim, J.-O. J.: The WRF single-moment 6-class microphysics scheme (WSM6), J. Korean Meteor. Soc, 42, 129-151, 2006.

Janić, Z. I.: Nonsingular implementation of the Mellor-Yamada level 2.5 scheme in the NCEP Meso model, US Department of Commerce, National Oceanic and Atmospheric Administration, National Weather Service, National Centers for Environmental Prediction, 2001.

Chen, F., and Dudhia, J.: Coupling an advanced land surface–hydrology model with the Penn State–NCAR MM5 modeling system. Part II: Preliminary model validation, Monthly Weather Review, 129, 587-604, 2001.

Emmons, L. K., Schwantes, R. H., Orlando, J. J., Tyndall, G., Kinnison, D., Lamarque, J.-F., et al., (2020). The Chemistry Mechanism in the Community Earth System Model version 2 (CESM2). Journal of Advances in Modeling Earth Systems, 12, e2019MS001882, https://doi.org/10.1029/2019MS001882

[*Table S1*]:

**Table S1** Model configuration for the simulation domain, meteorological schemes, chemical mechanisms, initial and lateral conditions, and emission inventories.

| Domain | |
| --- | --- |
| Size | 300 × 300 horizontal grid cells |
| Center | 116°E, 38° N |
| Horizontal resolution | 6 km × 6 km |
| Vertical resolution | 35 vertical levels, uneven intervals, spacing ranging from ~50 m near the surface, ~500 m at 2.5 km above the ground level, and more than 1 km at 14 km above the ground level |
| **Meteorology** | |
| Microphysics scheme | WSM 6-class grapple microphysics scheme (Hong and Lim, 2006) |
| Boundary layer scheme | MYJ PBL scheme (Janjić, 2002) |
| Surface layer scheme | Monin-Obukhov surface layer scheme (Janjić, 2002) |
| Land-surface scheme | Noah land-surface model (Chen and Dudhia, 2001) |
| Longwave radiation scheme | Goddard (Dudhia, 1989) |
| Shortwave radiation scheme | Goddard (Dudhia, 1989) |
| Dry deposition | Wesely (1989) |
| Wet deposition | CMAQ (Binkowski and Roselle, 2003) |
| **Chemistry** | |
| Gas phase chemistry | SAPRC99 chemical mechanism (Binkowski and Roselle, 2003) |
| Inorganic aerosols | ISORROPIA version 1.7 (Nenes et al., 1998) |

| | |
|---|---|
| Secondary organic aerosol | Nontraditional VBS parametrization (Li et al., 2011) |
| Photolysis rates | FTUV radiation transfer model (Tie et al., 2003) |
| **Boundary and initial conditions** | |
| Meteorological | NCEP FNL 6-hr 1° × 1° analysis data |
| Chemical | CAM-chem 6-hr outputs |
| **Emission inventory** | |
| Anthropogenic | MEIC (Zhang et al. 2009; Li et al., 2017) |
| Biogenic | MEGAN (Guenther et al., 2006) |
* * *
***Minor.2*** *Present percentage reductions in emissions during the lockdown to contextualize the observed PM$_{2.5}$ changes, enhancing the understanding of emission effectiveness.*

We revised the manuscript to include specific percentage reductions in emissions during the lockdown. Thank you.

**[Lines 185 in *Section 2.2*]:**

In the EMIS experiment, we used the anthropogenic emission inventory from the BASE case. Still, we excluded any abrupt decreases associated with anthropogenic emission reductions during the COVID-19 lockdown period 2020, following the provincial emission reduction ratios provided by Huang et al. (2021) (**Table S2**).

**[Table S2]:**

**Table S2** Provincial emission reduction ratios during the COVID-19 lockdown period in 2020 in the study area.

| Species / Province | CO | NO$_x$ | SO$_2$ | VOCs | PM$_{2.5}$ | BC | OC |
|---|---|---|---|---|---|---|---|
| Beijing | 22% | 45% | 26% | 45% | 18% | 46% | 8% |
| Tianjin | 21% | 38% | 20% | 41% | 14% | 22% | 6% |
| Hebei | 15% | 45% | 16% | 36% | 12% | 17% | 5% |
| Anhui | 14% | 56% | 22% | 31% | 11% | 22% | 4% |
| Inner Mongolia | 14% | 29% | 15% | 34% | 13% | 16% | 6% |
| Shaanxi | 19% | 45% | 18% | 34% | 13% | 22% | 5% |
| Hubei | 19% | 55% | 23% | 35% | 16% | 23% | 10% |

| | | | | | | | |
|---|---|---|---|---|---|---|---|
| Jilin | 16% | 39% | 23% | 34% | 13% | 18% | 5% |
| Liaoning | 21% | 40% | 28% | 36% | 16% | 28% | 8% |
| Henan | 23% | 57% | 22% | 41% | 18% | 35% | 8% |
| Shandong | 23% | 50% | 25% | 39% | 19% | 35% | 9% |
| Jiangsu | 23% | 50% | 26% | 41% | 16% | 35% | 7% |
| Shanghai | 35% | 48% | 42% | 45% | 34% | 54% | 42% |
* * *
***Minor.3*** *In section 3.1, the formulas from 1 to 3 are garbled, please correct them.*

Thank you for your comment. We have reviewed and corrected the formulas in Section 3.1. Additionally, these formulas have been moved to the Supplementary Material (Text S1) to improve clarity and organization.

**[Lines 128 in *Section 2.1*]:**

We validated the final emission inventory using statistical parameters, including normalized mean bias (*NMB*), index of agreement (*IOA*), and correlation coefficient (*r*) (**Text S1**).

**[Text S1]:**

**Text S1 Statistical methods for comparisons**

We assessed the model performance using several statistical parameters, including normalized mean bias (*NMB*), index of agreement (*IOA*), and correlation coefficient (*r*), to compare simulations against observational data. The evaluated variables encompass air pollutants such as $PM_{2.5}$, $O_3$, $NO_2$, $SO_2$, and CO concentrations within the NNCP and SNCP regions. $PM_{2.5}$ components, including organic, nitrate, sulfate, and ammonium, are also assessed at the IAP monitoring site. These statistical metrics provide a quantitative measure of how well the model reproduces the observed data, offering insights into its accuracy and reliability in simulating the atmospheric conditions and pollutant levels during the specified period.

$$NMB = \frac{\sum_{i=1}^{N}(P_i - O_i)}{\sum_{i=1}^{N} O_i} \tag{1}$$

$$IOA = 1 - \frac{\sum_{i=1}^{N}(P_i - O_i)^2}{\sum_{i=1}^{N}(|P_i - \overline{O}| + |O_i - \overline{O}|)^2} \tag{2}$$

$$r = \frac{\sum_{i=1}^{N}(P_i - \overline{P})(O_i - \overline{O})}{[\sum_{i=1}^{N}(P_i - \overline{P})^2 \sum_{i=1}^{N}(O_i - \overline{O})^2]^{\frac{1}{2}}} \tag{3}$$

where $P_i$ and $O_i$ represent the calculated and observed variables, respectively. $N$ stands for the total number of predictions for comparison, and $\overline{O}$ and $\overline{P}$ denote the

average observations and simulations, respectively. The *IOA* ranges from 0 to 1, where a value of 1 indicates perfect agreement between the predictions and observations. The *r* ranges from -1 to 1, 1 indicating perfect spatial consistency between the observations and predictions.
* * *
***Minor.4*** *Please standardize the subscript for PM2.5 in the manuscript.*

We have reviewed and standardized the subscript for $PM_{2.5}$ throughout the manuscript to ensure consistency. Thank you for your careful attention to detail.
* * *
***Minor.5*** *Coloured or marked text in \*.pdf manuscript file is not allowed. Please provide a clean version of \*pdf manuscript file (with black text) with the next revision.*

Changed as suggested. Thank you.

---

## Author Comment (AC3)

**Response to Reviewer #2**

*General Comments*

*The paper employed WRF-Chem to simulate PM2.5 formation in the North China Plain (NCP) during a lockdown period (January 21y 21 – February 16ry 16, 2020) under three scenarios: baseline, SEN_METEO, and SEN_EMIS. The SEN_METEO case replaced baseline meteorology with 2015-2019 mean climatology, while SEN_EMIS used baseline meteorology but substituted emissions with a no-lockdown scenario. By comparing their results, the study explores the impacts of meteorology and emission reductions on PM2.5 levels. Results indicate that, in the northern NCP, meteorological conditions had a stronger influence on PM2.5 levels than emission reductions, whereas, in the southern NCP, the benefits of emission reductions were more significant.*

Thank you for recognizing the critical components of our study. Your constructive feedback will significantly strengthen our manuscript. We respond to your concerns in detail below.

We respond to each specific comment in detail below. The reviewers' comments are shown in *black italics*. Our replies are in indented black text, and the modified text is in blue. The annotated line numbers refer to the revised copy of the manuscript.
* * *
*Major comments:*

*Overall, this study presents a solid approach with a well-evaluated model, but I have several concerns that need to be addressed before recommending this paper for publication:*

*Major.1 The title used "insights from 6-year simulations," but the manuscript appears to focus on one-month simulations for Jan-Feb 2020. It would be helpful to clarify the source of this "6-year" claim.*

Thank you for your constructive feedback. The phrase "insights from six-year simulations" in the original title was intended to highlight the climatological averages from 2015 to 2019, which provide a critical baseline for understanding the PM$_{2.5}$ dynamics during the one-month COVID-19 lockdown period. To address this and ensure clarity, we have revised the title and added detailed explanations throughout the manuscript.

**[*Title*]:**

"Impacts of meteorology and emission reductions on haze pollution during the Lockdown in the North China Plain"

**[Lines 189 in *Sect 2.2*]:**

In the METEO case, we applied the same emission inventory as the BASE case but with averaged meteorological conditions from 2015 to 2019. These mean meteorological fields were derived by averaging key meteorological variables (**Text S2**).

**[*Text S2*]:**

**Text S2 Mean meteorology from 2015 to 2019**

This study's mean meteorology field data was derived by averaging key meteorological variables (e.g., temperature, wind speed, relative humidity, and pressure) from 2015 to 2019. Given that the vertical levels in the NCEP FNL data varied across different years, we did not average the original data directly. Instead, we processed the data using the WRF Preprocessing System (WPS) to ensure consistency. Specifically, we ran WPS yearly to generate the met_em* files containing processed meteorological variables at uniform vertical levels and grid resolution. We then averaged these met_em* files across the six years at each grid point and pressure level, which helped preserve the atmospheric variables' vertical structure and physical coherence. This approach maintained a realistic representation of the atmospheric state by accounting for the multi-year variability while ensuring that the averaged fields were consistent with the WRF-Chem grid resolution. As the WPS processing already matched the data to the model's spatial resolution, no additional interpolation was required, thus ensuring the physical and spatial consistency of the averaged climatological fields used in the WRF-Chem simulations. This multi-year climatological averaging was designed to capture the typical variations in initial and boundary meteorological conditions. This approach provided a robust and representative baseline for multiple years, effectively minimizing the influence of anomalies or extreme weather events characteristic of any individual year.
* * *
*Major.2 In section 3.5, the discussion on "combined effects of meteorology and emission reduction" seems to involve a simple addition of the individual impacts of emissions and meteorology. This approach could be misleading. I suggest either comparing the magnitudes of these impacts separately or, if discussing combined*

*effects, perform a simulation that perturbs both emissions and meteorology simultaneously. Alternatively, you could have a separate section discussing how emission impacts vary under different meteorological conditions (EP1 vs. EP2 vs. non-haze episodes), as this question inherently addresses the coupled effects of emissions and meteorology.*

Thank you for your valuable suggestion. In response, we added a new simulation that simultaneously perturbs emissions and meteorological conditions (EMIS_METEO case). We also evaluated the combined and interactive effects of these factors more comprehensively. The data and analysis have been accordingly updated.

**[Lines 183 in *Sect 2.2*]:**

The other three groups are sensitivity simulations, which include the emission condition-sensitive simulation (EMIS), the meteorology condition-sensitive simulation (METEO), and the combined emission and meteorology condition-sensitive simulation (EMIS_METEO). In the EMIS experiment, we used the anthropogenic emission inventory from the BASE case. Still, we excluded any abrupt decreases associated with anthropogenic emission reductions during the COVID-19 lockdown period 2020, following the provincial emission reduction ratios provided by Huang et al. (2021) (**Table S2**). In the METEO case, we applied the same emission inventory as the BASE case but with averaged meteorological conditions from 2015 to 2019. These mean meteorological fields were derived by averaging key meteorological variables (**Text S2**). For the EMIS_METEO case, we used the emission inventory from the EMIS case and the mean meteorological conditions from the METEO case.

The comparison between the BASE and EMIS cases allowed us to evaluate the impact of sudden reductions in anthropogenic emissions on $PM_{2.5}$ levels. The comparison between the BASE and METEO cases provided a stable reference point by reducing the influence of anomalies or fluctuations in meteorological conditions from any year, enabling a comprehensive evaluation of the effects of meteorological factors on $PM_{2.5}$ levels. Finally, comparing the BASE and EMIS_METEO cases enabled a thorough assessment of the combined impact of emission reductions and meteorological conditions on $PM_{2.5}$ levels. Additionally, we analyzed the coupled effects between emission reductions and meteorological factors using a factor separation approach (**Text S3**).

**[Text S3]:**

**Text S3 Factor separation technique to analyze coupled effects**

In nonlinear atmospheric systems, factors often interact in complex ways, making it hard to identify their individual impacts. To address this, we used the factor separation approach (FSA) by Stein and Alpert (1993), which helps separate the direct effects of each factor from their interactions. In this study, we focused on emissions and meteorological changes, aiming to understand both their individual effects and how they interact. The pure contributions from emission reductions and meteorological changes are represented as follows:

$$f'_{EMIS} = f_{EMIS} - f_{BASE} \qquad (4)$$

$$f'_{METEO} = f_{METEO} - f_{BASE} \qquad (5)$$

When emissions and meteorological conditions are considered, the total impact includes their individual contributions and coupled. The combined effect is expressed as:

$$f_{EMIS\_METEO} = f'_{EMIS} + f'_{METEO} + f'_{EMIS\_METEO} + f_{BASE} \qquad (6)$$

To quantify the coupled effects between emissions and meteorological changes, we use the following equation:

$$f'_{EMIS\_METEO} = f_{EMIS\_METEO} - f'_{EMIS} - f'_{METEO} - f_{BASE}$$
$$= f_{EMIS\_METEO} - (f_{EMIS} - f_{BASE}) - (f_{METEO} - f_{BASE}) - f_{BASE}$$
$$= f_{EMIS\_METEO} - f_{EMIS} - f_{METEO} + f_{BASE} \qquad (7)$$

This final form helps us understand how the combined effects relate to individual impacts and the baseline. Using the FSA, we can clearly see how emissions and meteorological conditions contribute to changes in the atmosphere.

**[Lines 411 in *Sect. 3.5*]:**

[revised manuscript text omitted]

[**Figure S13**]

[Figure]

**Figure S13.** The coupled effects between emission reductions and meteorological factors on PM$_{2.5}$. The color gradient coupled effects averaged from (a) the entire study period, (b) the non-haze period, (c) the EP1 haze period, and (d) the EP2 haze period.

[**Figure 8**]

[Figure]

**Figure 8.** Regional contributions to PM$_{2.5}$ averaged in (a) the NNCP and (b) the SNCP during the entire period, non-haze period, EP1, and EP2. The contributions include meteorological conditions (METEO), abrupt anthropogenic emissions (EMIS) decreases, and coupled and combined effects of METEO and EMIS.

[*Figure S14*]

[Figure]

**Figure S14.** Regional contributions to daily PM$_{2.5}$ averaged in (a) the NNCP and (b) the SNCP. The contributions include meteorological conditions (METEO), abrupt decreases in anthropogenic emissions (EMIS), and synergistic effects of METEO and EMIS.
* * *
***Major.3*** *The study's novelty feels somewhat limited, as numerous previous studies have explored the relationships between emission reductions, meteorology, and air quality during the COVID-19 lockdown, some of which are referenced in this manuscript. The approach and findings do not seem to offer significant new insights or contradictions compared to existing literature. It would be helpful if the authors could more explicitly highlight the innovative aspects of their approach and clarify the novelty of their findings.*

Thank you for your valuable feedback on the novelty of our study. We have revised the manuscript to highlight our approach's unique contributions, particularly the methodologies used and the regional insights provided. Specifically, we clarified how using the WRF-Chem model with sensitivity experiments (e.g.,

SEN_METEO_EMIS) and factor separation methods offers a new perspective on the relationship between meteorology and emission reductions. Additionally, we emphasized our findings on the differing responses between the northern and southern North China Plain (NCP), which enhanced the understanding of air quality dynamics during the lockdown.

**[Lines 24 in *Abstract*]:**

Our analysis highlights a marked regional contrast: in the Northern NCP (NNCP), adverse meteorology largely offset emission reductions, resulting in $PM_{2.5}$ increases of 30 to 60 μg m$^{-3}$ during haze episodes. Conversely, the Southern NCP (SNCP) benefited from favourable meteorological conditions that lowered $PM_{2.5}$ by 20 to 40 μg m-3, combined with emission reductions. These findings emphasize the critical role of meteorology in shaping the air quality response to emission changes, particularly in regions like the NNCP, where unfavourable weather patterns can counteract the benefits of emission reductions. Our study provides valuable insights into the complex interplay of emissions, meteorology, and pollutant dynamics, suggesting that adequate air quality strategies must integrate emissions controls and meteorological considerations to address regional variations effectively.

**[Lines 79 in *Introduction*]:**

We emphasize the localized differences in how meteorological conditions and emission reductions affect air quality within the North China Plain, specifically between the Northern North China Plain (NNCP) and Southern North China Plain (SNCP). Utilizing the WRF-Chem model, we conducted detailed sensitivity experiments that allowed us to isolate and quantify the individual and combined impacts of emissions and meteorology on air quality, which can deepen the understanding of air quality dynamics in different regional contexts.

**[Lines 464 in *Conclusions*]:**

Previous studies have primarily focused on the overall impacts of meteorological conditions and emission reductions on air quality across the North China Plain and even nationwide. We emphasize the localized differences in how meteorological conditions and emission reductions affect air quality within the North China Plain, specifically between the NNCP and SNCP. Our findings underscore the critical role that meteorological conditions play in modulating the effects of emission reductions. The combination of unfavourable meteorological factors and emission reductions in the NNCP led to overall increases in $PM_{2.5}$ levels, with significant increases during haze episodes. Meanwhile, in the SNCP, meteorological conditions and emission reductions consistently contributed to lower $PM_{2.5}$ concentrations.
* * *
***Major.4*** *The clarity and logical flow of the manuscript could be improved, especially given the multiple sets of comparisons (e.g., SEN_EMIS vs. baseline, SEN_METEO vs. baseline, haze vs. non-haze, NNCP vs. SNCP). At times, these discussions get mixed, making it difficult to follow. For example, section 3.4 compares SEN_EMIS vs. baseline (with the same meteorology) but mentions "decreased atmospheric transport" (line 329), which is confusing – perhaps this refers to EP2 vs. other episodes? If the aim is to explore how emission impacts vary under different meteorological conditions, this should be clearly stated and organized into a separate section/paragraph. This issue appears elsewhere as well, and it would be helpful to clearly signal when switching between comparison sets.*

Thank you for your insightful feedback. We reorganized the Results and Discussion sections to improve the clarity and logical flow of the manuscript, explicitly addressing the need to separate discussions of meteorological and emission impacts. Section 3.4 has been revised to focus solely on the effects of emissions under constant meteorological conditions (EMIS vs. baseline). The reference to "decreased atmospheric transport", which was indeed confusing, has been clarified. This discussion now pertains to the combined and coupled effects between emissions and meteorology and has been moved to *Section 3.5*, where we discuss the newly added EMIS_SEN simulations and their interaction with meteorological conditions.
* * *
***Specific comments:***

***Specific.1*** *Page 5 line 96: How were the two regions of interest defined? Why are other parts of the NCP not included in your analysis or discussions?*

Thank you for the thoughtful comment. Strict geographical limits do not bind the delineation of the NNCP and SNCP; instead, it is based on representative features and differences critical for a comprehensive assessment of geographical, meteorological, and emission characteristics. The boundaries were drawn to effectively capture the distinct local attributes of each region, allowing for meaningful comparisons and insights into air quality dynamics. Regional differentiation is crucial for understanding the air quality dynamics across the NCP.

**[Lines 96 in *Section 2.1*]:**

We defined these regions by thoroughly analyzing geographical features, weather conditions, and emission sources. The NNCP, which generally includes the cities in the Beijing-Tianjin-Hebei (BTH) area, is surrounded by mountains and elevated terrain to the north and west. These features make it harder for pollutants to

disperse, leading to pollutant buildup, especially in winter when stagnant atmospheric conditions dominate (Feng et al., 2020; Li et al., 2019). On the other hand, the SNCP is characterized by lower elevations and broad plains, which help disperse pollutants due to more vital wind patterns and higher planetary boundary layer heights (Huang et al., 2021). The emissions in these two regions also differ significantly. The NNCP is mainly affected by concentrated urban and industrial emissions from the BTH area. At the same time, the SNCP has a broader variety of sources, including industrial and agricultural emissions, creating a more diverse pollutant profile(Zheng et al., 2021). These differences in geography, weather, and emissions provide a basis for studying how meteorological factors and emission reductions affect air quality differently across the NCP (**Figure 1**). By examining these sub-regions separately, we can better understand how air quality interventions vary in effectiveness across different areas.

* * *
***Specific.4*** *Page 6 line 134: You mentioned "6-year simulations" in the title, but this section states the simulations were conducted from January 21 to February 16, 2020. Does this mean they are one-month simulations only?*

Thank you for your constructive feedback. The phrase "insights from six-year simulations" in the original title was intended to highlight the climatological averages from 2015 to 2019, which provide a critical baseline for understanding the $PM_{2.5}$ dynamics during the one-month COVID-19 lockdown period. To address this and ensure clarity, we have revised the title and added detailed explanations throughout the manuscript.

**[*Title*]:**

"Impacts of meteorology and emission reductions on haze pollution during the Lockdown in the North China Plain"

**[Lines 189 in *Sect 2.2*]:**

In the METEO case, we applied the same emission inventory as the BASE case but with averaged meteorological conditions from 2015 to 2019. These mean meteorological fields were derived by averaging key meteorological variables (**Text S2**).

**[*Text S2*]:**

**Text S2 Mean meteorology from 2015 to 2019**

This study's mean meteorology field data was derived by averaging key meteorological variables (e.g., temperature, wind speed, relative humidity, and pressure) from 2015 to 2019. Given that the vertical levels in the NCEP FNL data varied across different years, we did not average the original data directly. Instead, we processed the data using the WRF Preprocessing System (WPS) to ensure consistency. Specifically, we ran WPS yearly to generate the met_em* files containing processed meteorological variables at uniform vertical levels and grid resolution. We then averaged these met_em* files across the six years at each grid point and pressure level, which helped preserve the atmospheric variables' vertical structure and physical coherence. This approach maintained a realistic representation of the atmospheric state by accounting for the multi-year variability while ensuring that the averaged fields were consistent with the WRF-Chem grid resolution. As the WPS processing already matched the data to the model's spatial resolution, no additional interpolation was required, thus ensuring the physical and spatial consistency of the averaged climatological fields used in the WRF-Chem simulations. This multi-year climatological averaging was designed to capture the typical variations in initial and boundary meteorological conditions. This approach provided a robust and representative baseline for multiple years, effectively minimizing the influence of anomalies or extreme weather events characteristic of any individual year.
* * *
***Specific.5*** *Page 7 line 152: It would be useful to elaborate on how the climatology was averaged. Did you average all meteorological variables directly? If so, how did you ensure the averaged climatology remained physically coherent? Was interpolation done to match the WRF-Chem grid resolution?*

Thank you for your insightful questions. We provided more detailed processing steps, explaining our approach to creating a physically coherent climatology for the simulations.

**[Lines 189 in *Sect 2.2*]:**

In the METEO case, we applied the same emission inventory as the BASE case but with averaged meteorological conditions from 2015 to 2019. These mean meteorological fields were derived by averaging key meteorological variables (**Text S2**).

**[*Text S2*]:**

**Text S2 Mean meteorology from 2015 to 2019**

This study's mean meteorology field data was derived by averaging key meteorological variables (e.g., temperature, wind speed, relative humidity, and pressure) from 2015 to 2019. Given that the vertical levels in the NCEP FNL data varied across different years, we did not average the original data directly. Instead, we processed the data using the WRF Preprocessing System (WPS) to ensure consistency. Specifically, we ran WPS yearly to generate the met_em* files containing processed meteorological variables at uniform vertical levels and grid resolution. We then averaged these met_em* files across the six years at each grid point and pressure level, which helped preserve the atmospheric variables' vertical structure and physical coherence. This approach maintained a realistic representation of the atmospheric state by accounting for the multi-year variability while ensuring that the averaged fields were consistent with the WRF-Chem grid resolution. As the WPS processing already matched the data to the model's spatial resolution, no additional interpolation was required, thus ensuring the physical and spatial consistency of the averaged climatological fields used in the WRF-Chem simulations. This multi-year climatological averaging was designed to capture the typical variations in initial and boundary meteorological conditions. This approach provided a robust and representative baseline for multiple years, effectively minimizing the influence of anomalies or extreme weather events characteristic of any individual year.
* * *
*Specific.6    Page 10 line 228: The exact time periods for EP1 and EP2 should be clearly stated here.*

Thank you for your comment. We included the specific dates for each episode in the manuscript.

**[Lines 264 in *Sect. 3.2*]:**

During the study period, two significant haze episodes were identified: EP1, lasting from January 22 to 29, and EP2, from February 8 to 13.
* * *
***Specific.7*** *Page 10 line 233: Since Figures 5-7 show "non-haze times," it would be helpful to explain the atmospheric conditions during those periods as well.*

Thank you for the insightful suggestion. We added a comprehensive overview of the atmospheric conditions during non-haze periods.

**[Lines 329 in *Sect. 3.3*]:**

During non-haze periods, weather conditions still significantly impacted $PM_{2.5}$ levels across the region, though the effect was less intense than haze episodes. In the NNCP, stagnant air and low wind speeds led to $PM_{2.5}$ increases of 10 to 30 µg m$^{-3}$ (**Figure 5b**). These weak conditions prevented effective pollutant dispersion, causing pollutants to accumulate, although less than during significant pollution events. This ongoing buildup due to poor weather shows the continued vulnerability of the NNCP to limited ventilation (Feng et al., 2021; Yan et al., 2024). In contrast, in the SNCP, weather conditions helped reduce $PM_{2.5}$ by 10 to 30 µg m$^{-3}$ (**Figure 5b**). This improvement was mainly due to higher PBLH (**Figure S7b**) and stronger winds (**Figure 5b**), which promoted pollutant dispersion. The PBLH rose by 100 to 300 meters, allowing pollutants to spread vertically, leading to lower $PM_{2.5}$ levels at the surface. Favorable winds also helped clear pollutants, enhancing the positive effects of meteorology on air quality. Previous studies have shown that regions with better dispersion conditions can achieve more significant air quality improvements, even with similar emissions, due to more efficient pollutant removal (Xu et al., 2020b; Zhang et al., 2021). These regional differences during non-haze periods show the critical role of weather in influencing air quality. In the NNCP, weak atmospheric circulation limited pollutant dispersion, causing moderate $PM_{2.5}$ increases. In contrast, in the SNCP, more dynamic weather conditions promoted pollutant removal, leading to substantial reductions.

**[*References*]**

Feng J, Liao H, Li Y, et al. Long-term trends and variations in haze-related weather conditions in north China during 1980–2018 based on emission-weighted stagnation intensity[J]. Atmospheric Environment, 2020, 240: 117830.

Yan, F., Su, H., Cheng, Y., Huang, R., Liao, H., Yang, T., Zhu, Y., Zhang, S., Sheng, L., Kou, W., Zeng, X., Xiang, S., Yao, X., Gao, H., and Gao, Y.: Frequent haze events associated with transport and stagnation over the corridor between the North China Plain and Yangtze River Delta, Atmos. Chem. Phys., 24, 2365–2376, https://doi.org/10.5194/acp-24-2365-2024, 2024.

Xu Y, Xue W, Lei Y, et al. Spatiotemporal variation in the impact of meteorological conditions on PM2. 5 pollution in China from 2000 to 2017[J]. Atmospheric Environment, 2020, 223: 117215.

Zhang, S., Zeng, G., Yang, X., Wu, R., and Yin, Z.: Comparison of the influence of two types of cold surge on haze dispersion in eastern China, Atmos. Chem. Phys., 21, 15185–15197, https://doi.org/10.5194/acp-21-15185-2021, 2021.
* * *
***Specific.8*** *Page 12 line 303: Have you examined the impact of meteorological conditions on biogenic emissions? If so, what role does it play?*

Thank you for the insightful question. During the winter months, biogenic emissions are limited due to lower temperatures, which reduce the release of biogenic volatile organic compounds (BVOCs). Therefore, the overall contribution of biogenic secondary organic aerosols (BSOAs) to PM2.5 concentrations is minimal during this period. We have clarified this point in the revised text and provided supporting data to show that the BSOA contribution is less than 2 µg/m³, representing less than 2% of total PM2.5 concentrations during the study period.

**[Lines 368 in *Sect. 3.3*]:**

Due to the very low emissions of biogenic secondary organic aerosol (BSOA) precursors during wintertime(Guenther et al., 2012), the BSOA contribution to $PM_{2.5}$ concentrations is insignificant, averaging less than 2 µg m$^{-3}$ throughout the study period (**Figure S11a**). The average BSOA accounted for less than 2% of total $PM_{2.5}$ mass in the BASE simulations (**Figure S11b**), indicating a minor role for biogenic emissions in shaping wintertime air quality.

**[*References*]**

Guenther A B, Jiang X, Heald C L, et al. The Model of Emissions of Gases and Aerosols from Nature version 2.1 (MEGAN2. 1): an extended and updated framework for modeling biogenic emissions[J]. Geoscientific Model Development, 2012, 5(6): 1471-1492.

**[*Figure S11*]:**

[Figure]

**Figure S11.** Spatial distribution of (a) near-surface biogenic SOA mass concentration and (b) its contribution as a percentage of PM$_{2.5}$ in the BASE simulations over the study period.
* * *
***Specific.9*** *Figure 4: Clarify what "all time" refers to. Does it mean the one-month period (January 21y 21 to February 16ry 16, 2020) or the 6-year period mentioned in the title?*

Thank you for your valuable observation. We replaced "all time" with "the study period" (January 21 to February 16, 2020). This change has been reflected in the relevant figures (Figs. 4-7 and Figs. S8, S13) to avoid confusion and ensure consistency.
* * *
***Specific.10*** *Figure 5-8: Typically, anomaly values are calculated as [scenario X minus baseline]. If your figures show [baseline minus scenario X], it would be helpful to explicitly mention this in the legend to avoid confusion.*

We explicitly indicated in the figure legends that the values displayed represent [baseline minus scenario X].

**Figure 5.** The pattern comparisons of the "BASE" simulation minus the "METEO" simulation. The color gradient represents PM2.5 changes averaged from (a) the entire study period, (b) the non-haze period, (c) the EP1 haze period, and (d) the EP2 haze period, along with the simulated surface wind fields.

**Figure 6.** The pattern comparisons of the "BASE" simulation minus the "EMIS" simulation. The color gradient represents $PM_{2.5}$ changes averaged from (a) the entire study period, (b) the non-haze period, (c) the EP1 haze period, and (d) the EP2 haze period.

**Figure 7.** The pattern comparisons of the "BASE" simulation minus the "EMIS_METEO" simulation. The color gradient represents coupled effects on $PM_{2.5}$ averaged from (a) the entire study period, (b) the non-haze period, (c) the EP1 haze period, and (d) the EP2 haze period.

**Figure S8.** The pattern comparisons of the "BASE" simulation minus the "METEO" simulation. The color gradient represents PBLH changes averaged from (a) the entire study period, (b) the non-haze period, (c) the EP1 haze period, and (d) the EP2 haze period.

**Figure S13.** The coupled effects between emission reductions and meteorological factors on $PM_{2.5}$. The color gradient coupled effects averaged from (a) the entire study period, (b) the non-haze period, (c) the EP1 haze period, and (d) the EP2 haze period.
* * *
***Specific.11*** *Figure 9: Refer to my major comment 2. The calculation of "combined effects" by simply adding meteorological and emission impacts is misleading.*

Thank you for your valuable suggestion. We introduced a new simulation case (EMIS_METEO) that simultaneously perturbs emissions and meteorological conditions. This simulation allows us to assess these two factors' combined and coupled effects comprehensively. The data and analysis have been updated to reflect these changes.

Please refer to our detailed explanation in the response to Major Comment 2, where we elaborate on these updates and their implications for our findings.
* * *
***Technical corrections:***

***Technical.1*** *Page 3 line 67: "… haze above event" --> "… above haze event"*

Changed as suggested. Thank you.
* * *
***Technical.2*** *Page 4 line 74: Duplicate citations*

We carefully reviewed the manuscript to remove any duplicate references. Thank you.
* * *
***Technical.3*** *Page 5 line 101: "PM2.5, O3, NO2, SO2 and CO" --> "PM2.5, O3, NO2, SO2 and CO"; check subscript formatting throughout the manuscript.*

We carefully reviewed the manuscript to correct any subscript formatting. Thank you.
* * *
***Technical.4*** *Page 6 line 136: Rephrase "consisted of a grid of 300 by 300 points, each spaced at a resolution of 6km" to "consisted of 300 × 300 horizontal grid cells with a 6 km resolution"*

Changed as suggested. Thank you.
* * *
***Technical.5*** *Page 6 line 139: Define the acronym "NCDP FNL" when first introduced*

Changed as suggested. Thank you.

**[Lines 168 in *Sect. 2.2*]:**

"the National Centers for Environmental Prediction (NCEP) Final (FNL)"